



# Evaluated kinetic and photochemical data for atmospheric chemistry: Volume VIII - gas phase reactions of organic species with four, or more, carbon atoms (≥ C4)

Abdelwahid Mellouki[1], Markus Ammann[2], R. Anthony Cox[3], John N. Crowley[4], Hartmut Herrmann[5], Michael E. Jenkin[6], V. Faye McNeill[7], Jürgen Troe[8,9], and Timothy J. Wallington[10]

[1] ICARE-CNRS, 1 C Av. de la Recherche Scientifique, 45071 Orléans Cedex 2, France
[2] Laboratory of Radiochemistry and Environmental Chemistry, OFLB 103, Paul Scherrer Institut, 5232 Villigen, Switzerland
[3] Centre for Atmospheric Science, Dept. of Chemistry, University of Cambridge, Lensfield Road, Cambridge CB2 1EP, UK
[4] Max Planck Institute for Chemistry, Division of Atmospheric Chemistry, 55128 Mainz, Germany
[5] Leibniz Institute for Tropospheric Research (TROPOS), Atmospheric Chemistry Dept. (ACD), 04318 Leipzig, Germany
[6] Atmospheric Chemistry Services, Okehampton, Devon, EX20 4QB, UK
[7] Department of Chemical Engineering, Columbia University, New York, NY 10027, USA
[8] Max Planck Institute for Biophysical Chemistry, Am Fassberg 11, 37077 Göttingen, Germany
[9] Institute for Physical Chemistry, University of Göttingen, Tammannstr. 6, 37077 Göttingen, Germany
[10] Ford Motor Company, Research and Advanced Engineering, Mail Drop RIC-2122, Dearborn, Michigan 48121-2053, USA

*Correspondence to*: A. Mellouki (mellouki@cnrs-orleans.fr)

**Abstract.** This article, the eighth in the series, presents kinetic and photochemical datasheets evaluated by the IUPAC Task Group on Atmospheric Chemical Kinetic Data Evaluation. It covers the gas phase thermal and photochemical reactions of organic species with four, or more, carbon atoms (≥ C4) available on the IUPAC website in 2019, including thermal reactions of closed-shell organic species with HO and NO3 radicals, and their photolysis. The article consists of a summary table, containing the recommended kinetic parameters for the evaluated reactions, and a supplement containing the datasheets, which provide information upon which the recommendations are made.

## 1 Introduction

In the mid-1970s it was appreciated that there was a need for the establishment of an international panel to produce a set of critically evaluated rate parameters for reactions of interest for atmospheric chemistry. To this end the CODATA Task Group on Chemical Kinetics, under the auspices of the International Council of Scientific Unions (ICSU), was constituted in 1977, and tasked to produce an evaluation of relevant, available kinetic and photochemical data. The first evaluation by this international committee was published in J. Phys. Chem. Ref. Data



in 1980 (Baulch et al., 1980), followed by Supplements in 1982 (Baulch et al., 1982) and 1984 (Baulch et al., 1984). In 1986 the IUPAC Subcommittee on Gas Kinetic Data Evaluation for Atmospheric Chemistry superseded the original CODATA Task Group for Atmospheric Chemistry. The Subcommittee continued its data evaluation program with Supplements published in 1989 (Atkinson et al., 1989), 1992 (Atkinson et al., 1992), 1997 (Atkinson et al.,

1997a), 1997 (Atkinson et al., 1997b), 1999 (Atkinson et al., 1999), and 2000 (Atkinson et al., 2000).

The gas-phase evaluation work was expanded to include heterogeneous reactions of gases on solid and liquid substrates in 2005. Aqueous-phase reactions of atmospheric importance were

added in 2015. The IUPAC group's work now includes over 1400 gas-phase, heterogeneous, and aqueous-phase reactions of importance in atmospheric chemistry. Reflecting the broader scope the group changed its name to the IUPAC Task Group on Atmospheric Chemical Kinetic Data Evaluation in 2013. Cox (2012) and Cox et al. (2018) discuss the history of IUPAC data evaluations and their role in addressing the critical societal challenges of stratospheric ozone

loss, tropospheric ozone formation, acid rain, urban air pollution, aerosol formation, and climate change.

In 2000 the evaluation was made available on the worldwide web (since 2016, http://iupac.pole-ether.fr). The IUPAC website hosts an interactive data base with a search facility and hyperlinks

between the summary table and the datasheets, both of which can be downloaded as individual Word and PDF files. Work is underway to convert the datasheets to machine readable xml files which will enable automatic transfer of IUPAC recommended data into atmospheric models. The IUPAC group continues to update and extend the set of evaluated reactions. To enhance the accessibility of this updated material to the scientific community, the evaluation is being

published as a series of articles in Atmospheric Chemistry and Physics (Atkinson et al., 2004, 2006, 2007, 2008; Crowley et al., 2010; Ammann et al., 2013). We present here in Volume VIII new datasheets for gas-phase thermal and photochemical initiation reactions of organic species with four, or more, carbon atoms. The coverage of this volume includes evaluation of the thermal reactions of the organic species with HO and $NO_3$ radicals, and photolysis. The

reactions with $O_3$ are included in a separate volume within this series (Cox et al., 2020), along with the chemistry of the Criegee intermediates produced.





## 2 Guide to the gas-phase datasheets

For each reaction covered in this volume, a datasheet with details about e.g. experimental methods and a justification of the choice of preferred value is available in the supplementary information. The datasheets covering gas-phase reactions are principally of two types: (i) those for individual thermal reactions and (ii) those for the individual photochemical reactions.

**2.1 Thermal reactions**

The datasheets begin with a statement of the reactions including known / potential product channels when this information is available. The available kinetic data on the reactions are summarized under two headings: (i) Absolute Rate Coefficients, and (ii) Relative Rate

Coefficients. Under both headings, we list the published experimental data as absolute rate coefficients. If the temperature coefficient has been measured, the results are given in a temperature dependent form over a stated temperature range. For bimolecular reactions, the temperature dependence is usually expressed in the normal Arrhenius form, $k = A \exp(-B/T)$, where B = E/R. For a few bimolecular reactions, we have listed temperature dependencies in

alternative forms such as $k = C(T/298 \text{ K})^n \exp(-D/T)$ or $k = ET^2\exp(-F/T)$ where the original authors have found that alternative expressions give a better fit to the data. In our recommendations we seek to provide simple Arrhenius expressions that describe the kinetics over the atmospherically relevant temperature range (200-300 K). More complex expressions which are often needed to describe the kinetic behaviour over larger ranges of temperature are

given in the Comments on Preferred Values section in the datasheets. Rate coefficients are given here in units of $cm^3$ molecule$^{-1}$ s$^{-1}$. Note that "molecule" is not a unit, but is included for clarity. For pressure dependent combination and dissociation reactions, the non-Arrhenius temperature dependence is used. This is discussed more fully in a subsequent section of this guide. Single temperature data are presented as such and wherever possible the rate coefficient at, or close to,

298 K is quoted directly as measured by the original authors. This means that the listed rate coefficient at 298 K may differ slightly from that calculated from the Arrhenius parameters determined by the same authors. Rate coefficients at 298 K marked with an asterisk indicate that the value was calculated by extrapolation of a measured temperature range which did not include 298 K. The tables of experimental data are supplemented by a series of comments

summarizing the experimental details. The following abbreviations, relating to experimental techniques, are used in the Techniques and Comments sections:



A– absorption

AS – absorption spectroscopy

CCD – charge coupled detector

CIMS – chemical ionization mass spectroscopy/spectrometry

CL – chemiluminescence

CRDS – cavity ring-down spectroscopy

DF – discharge flow

EPR – electron paramagnetic resonance

F – flow system

FP – flash photolysis

FTIR – Fourier transform infrared

FTS – Fourier transform spectroscopy

GC – gas chromatography/gas chromatographic

HPLC – high-performance liquid chromatography

IR – infrared

LIF – laser induced fluorescence

LMR – laser magnetic resonance

LP – laser photolysis

MM – molecular modulation

MS – mass spectrometry/mass spectrometric

P – steady state photolysis

PLP – pulsed laser photolysis

PR – pulse radiolysis

RA – resonance absorption

RF – resonance fluorescence

RR – relative rate

S – static system

TDLS – tunable diode laser spectroscopy

UV – ultraviolet

UVA – ultraviolet absorption

VUVA – vacuum ultraviolet absorption

For measurements of relative rate coefficients, wherever possible the comments contain the



actual measured ratio of rate coefficients together with the rate coefficient of the reference reaction used to calculate the absolute rate coefficient listed in the data table. The absolute value of the rate coefficient given in the table may be different from that reported by the original author owing to a different choice of rate coefficient of the reference reaction. Whenever

possible the reference rate coefficient data are those preferred in the present evaluation.

The preferred values in the datasheets are based on our consideration of the suitability of experimental method and coverage of applicable parameter space (temperature, total pressure of diluent gas, partial pressure of gas-phase species) within the atmospherically relevant range.

The general approach and methods used have been reviewed recently by Cox (2012). It is recognized that preferred values may change with publication of new data, and such changes are updated at the website. The preferred rate coefficients are presented (i) at a temperature of 298 K and (ii) in temperature dependent form over a stated temperature range. This is followed by a statement of the uncertainty limits in log $k$ at 298 K and the uncertainty limits either in

(E/R) or in n, for the mean temperature in the range. Some comments on the assignment of uncertainties are given later in this guide to the datasheets. The Comments on Preferred Values describe how the selection was made and give any other relevant information. The extent of the comments depends upon the present state of our knowledge of the particular reaction in question. The datasheets are concluded with a list of the relevant references.


## 2.2 Conventions concerning rate coefficients

All of the reactions in the table are elementary processes. Thus the rate expression is derived from a statement of the reaction, e.g.


$$A + A \rightarrow B + C$$

$$-\frac{1}{2}\frac{d[A]}{dt} = \frac{d[B]}{dt} = \frac{d[C]}{dt} = k[A]^2 \qquad\qquad \text{Eq. (1)}$$

Note that the stoichiometric coefficient for A, i.e. 2, appears in the denominator before the rate of change of [A] (which is equal to $2k[A]^2$) and as a power on the right-hand side. Representations of $k$ as a function of temperature characterize simple "direct" bimolecular reactions. Sometimes it is found that $k$ also depends on the pressure and the nature of the bath



gas. This may be an indication of complex-formation during the course of the bimolecular
reaction, which is always the case in combination reactions. In the following sections the
representations of $k$ which are adopted in these cases are explained.

### 2.3 Treatment of combination and dissociation reactions

Unlike simple bimolecular reactions such as those considered in Sect. 2.2, combination
reactions

$$A + B + M \rightarrow AB + M$$

and the reverse dissociation reactions

$$AB + M \rightarrow A + B + M$$

are composed of sequences of different types of physical and chemical elementary processes.
Their rate coefficients reflect the more complicated sequential mechanism and depend on the
temperature, $T$, and the nature and concentration of the third body, M. In this evaluation, the
combination reactions are described by a formal second-order rate law:

$$\frac{\mathrm{d[AB]}}{\mathrm{d}t} = k[A][B] \qquad\qquad \text{Eq. (2)}$$


while dissociation reactions are described by a formal first-order rate law:

$$\frac{-\mathrm{d[AB]}}{\mathrm{d}t} = k[AB] \qquad\qquad \text{Eq. (3)}$$

In both cases, $k$ depends on the temperature and on the concentration of M, i.e., [M]. To
rationalize the representations of the rate coefficients used in this evaluation, we first consider
the Lindemann-Hinshelwood reaction scheme. The combination reactions follow an elementary
mechanism of the form,

$$A + B \rightarrow AB* \qquad\qquad (1)$$
$$AB* \rightarrow A + B \qquad\qquad (-1)$$
$$AB* + M \rightarrow AB + M \qquad\qquad (2)$$


while the dissociation reactions are characterized by:

$$AB + M \rightarrow AB^* + M \qquad\qquad (-2)$$

$$AB^* + M \rightarrow AB + M \qquad\qquad (2)$$

$$AB^* \rightarrow A + B \qquad\qquad (-1)$$

Assuming quasi-stationary concentrations for the highly excited unstable species AB* (i.e. that $d[AB^*]/dt \approx 0$), it follows that the rate coefficient for the combination reaction is given by:

$$k = k_1 \left( \frac{k_2[M]}{k_{-1}+k_2[M]} \right) \qquad\qquad \text{Eq. (4)}$$


while that for the dissociation reaction is given by:

$$k = k_{-2}[M] \left( \frac{k_{-1}}{k_{-1}+k_2[M]} \right) \qquad\qquad \text{Eq. (5)}$$

In these equations the expressions before the parentheses represent the rate coefficients of the process initiating the reaction, whereas the expressions within the parentheses denote the fraction of reaction events which, after initiation, complete the reaction to products. In the low pressure limit ($[M] \rightarrow 0$) the rate coefficients are proportional to $[M]$; in the high pressure limit ($[M] \rightarrow \infty$) they are independent of $[M]$. It is useful to express $k$ in terms of the limiting low

pressure and high pressure rate coefficients,

$$k_0 = \lim k([M]) \text{ for } [M] \rightarrow 0 \text{ and } k_\infty = \lim k([M]) \text{ for } [M] \rightarrow \infty \qquad\qquad \text{Eq. (6)}$$

From this convention, the Lindemann-Hinshelwood equation is obtained

$$k = \frac{k_0 k_\infty}{k_0 + k_\infty} \qquad\qquad \text{Eq. (7)}$$

It follows that, for combination reactions, $k_0 = k_1 k_2[M] / k_{-1}$ and $k_\infty = k_1$, while, for dissociation

reactions, $k_0 = k_{-2}[M]$ and $k_\infty = k_{-1}k_{-2} / k_2$. Since detailed balancing applies, the ratio of the rate coefficients for combination and dissociation at a fixed $T$ and $[M]$ is given by the equilibrium constant $K_c = k_1 k_2 / k_{-1}k_{-2}$.

Starting from the high-pressure limit, the rate coefficients fall off with decreasing third body



concentration [M] and the corresponding representation of $k$ as a function of [M] is termed the
"falloff curve" of the reaction. In practice, the above Lindemann-Hinshelwood expressions do
not suffice to characterize the falloff curves completely. Because of the multistep character of
the collisional deactivation ($k_2$[M]) and activation ($k_{-2}$[M]) processes, and energy- and angular
momentum-dependences of the association ($k_1$) and dissociation ($k_{-1}$) steps, as well as other
phenomena, the falloff expressions have to be modified. This can be done by including a
broadening factor $F$ to the Lindemann-Hinshelwood expression (Troe, 1979):

$$k = \frac{k_0 k_\infty}{k_0 + k_\infty} F = k_0 \left(\frac{1}{1+x}\right) F = k_\infty \left(\frac{x}{1+x}\right) F \qquad \text{Eq. (8)}$$

The broadening factor $F$ depends on the ratio $x = k_0/k_\infty$, which is proportional to [M], and can
be used as a measure of "reduced pressure". The first factors on the right-hand side represent
the Lindemann-Hinshelwood expression and the additional broadening factor $F$, at not too high
temperatures, is approximately given by (Troe, 1979):

$$\log F \cong \frac{\log F_c}{1 + [\log(k_0/k_\infty)/N]^2} \qquad \text{Eq. (9)}$$


where $\log = \log_{10}$ and $N \approx [0.75 - 1.27 \log F_c]$.

When $F_c$ decreases, the falloff curve broadens and becomes asymmetric (i.e. $F(k_0/k_\infty) \neq F(k_\infty/k_0)$). The given equation for $F$ then becomes insufficient and should be replaced, e.g. by


$$F(x) \approx (1+x)/(1+x^n)^{1/n} \qquad \text{Eq. (10)}$$

where $x = k_0/k_\infty$, $n = [\ln 2 / \ln(2/F_c)] [0.8 + 0.2 \, x^q]$, $q = (F_c - 1) / \ln(F_c/10)$ and $\ln = \log_e$ (Troe and
Ushakov, 2014). While the former equation for $\log F$ appears acceptable as long as $F_c \geq 0.6$,
the latter equation for $F$ should be used for $F_c \leq 0.6$. With these equations, falloff curves are
represented in terms of the three parameters $k_0$ (being proportional to [M]), $k_\infty$, and $F_c$.
The parameters $k_0$, $k_\infty$, and $F_c$ depend on details of the intra- and intermolecular dynamics and
in principle can be calculated. If the required information is not available, one has to obtain
them by fitting experimental falloff curves with the expressions given above. Nevertheless, one
may estimate $F_c$ to be typically of the order of 0.49, 0.44, 0.39, and 0.35, if the reactants A and



B in total have $r$ = 3, 4, 5, and 6 external rotational degrees of freedom, respectively (Cobos and Troe, 2003; for the reaction HO + NO$_2$ + M , e.g. one would have $r$ = 5 and $F_c \approx 0.39$); $F_c$ may be lower, if low frequency vibrations in A or B are relevant in addition to the rotations and if collisions are inefficient. Over the range 200 – 300 K often one can neglect a temperature

dependence of $F_c$ (for detailed calculations of $F_c$, including a dependence on the bath gas M, see e.g. Troe 1983; Troe and Ushakov, 2011, 2014). The accuracy of $F(x)$ as given above is estimated to be about 10 percent. Larger differences between experimentally fitted $F_c$ often are an indication for inadequate falloff extrapolations to $k_0$ and/or $k_\infty$. In this case, the apparent values for $k_0$, $k_\infty$, and $F_c$ still can provide a satisfactory representation of the considered

experimental data, in spite of the fact that $k_0$ and/or $k_\infty$ are not the real limiting values. If falloff curves are fitted in different ways, changes in $F_c$ require changes in the limiting $k_0$ and $k_\infty$. In the present evaluation, we generally follow the experimentally fitted values for $k_0$, $k_\infty$, and $F_c$, provided that $F_c$ does not differ too much from the standard values given above and theoretically modelled values. If large deviations are encountered, the experimental data are re-evaluated

using $F_c$ -values as given above. Values of $k_\infty$ for combination reactions without a barrier often have only weak temperature dependences which in practice can be neglected.

Besides the energy-transfer mechanism, i.e., reactions (1), (-1), and (2), a second mechanism may become relevant for some reactions considered here. This is the radical-complex (or

chaperon) mechanism

$$A + M \rightarrow AM \qquad\qquad (3)$$
$$AM \rightarrow A + M \qquad\qquad (-3)$$
$$B + AM \rightarrow AB + M \qquad\qquad (4)$$

which, in the low pressure range, leads to $k_0$ = $(k_3 / k_{-3})k_4$ [M]. For some tri- and tetra-atomic

adducts AB, e.g., O + O$_2$ $\rightarrow$ O$_3$ and HO + C$_6$H$_6$ $\rightarrow$ HOC$_6$H$_6$, the value of $k_0$ may exceed that from the energy-transfer mechanism and show stronger temperature dependences (Luther et al., 2005; Teplukhin and Babikov, 2016). This mechanism may also influence high pressure experiments when $k_0$ from the radical-complex mechanism exceeds $k_\infty$ from the energy-transfer mechanism (Oum et al., 2003). In this case falloff over wide pressure ranges cannot be

represented by contributions from the energy-transfer mechanism alone, in particular when measurements at pressures above about 10 bar are taken into consideration.

The dependence of $k_0$ and $k_\infty$ on the temperature $T$ is represented in the form $k \propto T^{-n}$ except for





cases with an established energy barrier in the potential. We have used this form of temperature

dependence because it usually gives a better fit to the data over a wider range of temperature than does the Arrhenius expression. It should be emphasised that the chosen form of the temperature dependence is often only adequate over limited temperature ranges such as 200– 300 K. Obviously, the relevant values of $n$ are different for $k_0$ and $k_\infty$. In this evaluation, values of $k_0$ are given for selected examples of third bodies M, and if possible for M = $N_2$, $O_2$, or air.


### 2.4 Treatment of complex-forming bimolecular reactions

Bimolecular reactions may follow the "direct" pathway

$$A + B \rightarrow C + D$$

and/or involve complex-formation, in the simplest way characterized by the steps

$$A + B \rightarrow AB^* \tag{1}$$


$$AB^* \rightarrow A + B \tag{-1}$$

$$AB^* \rightarrow C + D \tag{5}$$

$$AB^* + M \rightarrow AB + M \tag{2}$$

(there may be additional pathways following from AB*; direct and complex-forming pathways may or may not be coupled). Assuming quasi-stationary concentrations of AB* (i.e. that $d[AB^*]/dt \approx 0$ as in section 2.3), a Lindemann-Hinshelwood type analysis leads to


$$d[AB]/dt = k_{Ass} [A] [B] \qquad \text{Eq. (11)}$$

$$d[C]/dt = d[D]/dt = k_{CA} [A] [B] \qquad \text{Eq. (12)}$$

$$d[A]/dt = - ( k_{Ass} + k_{CA} ) [A] [B] \qquad \text{Eq. (13)}$$





The rate constants for association ($k_{Ass}$) and for chemical activation leading to product formation ($k_{CA}$) then are given by

$$k_{Ass} = k_1\, k_2\, [M]\, /(\, k_{-1} + k_2\, [M] + k_5) \qquad\qquad\qquad \text{Eq. (14)}$$

$$k_{CA} = k_1\, k_5\, /(\, k_{-1} + k_2\, [M] + k_5) \qquad\qquad\qquad \text{Eq. (15)}$$

Note that $k_{Ass}$ and $k_{CA}$ are dependent on the nature and concentration of the third body M, in
addition to their temperature dependence. In reality, as for combination and dissociation reactions, the given expressions for $k_{Ass}$ and $k_{CA}$ have to be extended by suitable broadening factors $F$ to account for the multistep character of processes (2) and the energy- and angular momentum-dependences of processes (1), (-1), and (5). These broadening factors, however, generally differ for $k_{Ass}$ and $k_{CA}$; also they generally differ from those of simple combination
reactions described in section 2.3. One should note that association and chemical activation here are coupled such that their joint treatment is complicated. Some simplification is reached when the processes first are treated separately and the coupling is introduced at the end (Troe, 2015). The corresponding rate constants of the separated processes are denoted by $k_{Ass}*$ and $k_{CA}*$ and are given by

$$k_{Ass}* = k_1\, k_2\, [M]\, /\, (k_{-1} + k_2\, [M]) \qquad\qquad\qquad \text{Eq. (16)}$$

and

$$k_{CA}* = k_1\, k_5\, /\, (k_2\, [M] + k_5). \qquad\qquad\qquad \text{Eq. (17)}$$

$k_{Ass}*$ then corresponds to the rate constant of a combination reaction described in section 2.3 and has a broadening factor $F_{Ass}*(x*)$. $k_{CA}*$ has to be treated in a different way and is expressed in the form

$$k_{CA}* = k_{Ass,\infty}\, [\, 1/\, (1 + x*)]\, F_{CA}*(x^*) \qquad\qquad\qquad \text{Eq. (18)}$$

with $x^* = k_{Ass,\infty}\, [M]\, /\, k_{CA,\infty}*$ and a broadening factor $F_{CA}*(x)$ (Stewart et al., 1989). The latter factor is generally larger than $F_{Ass}*(x*)$ (Troe, 2015). The rate parameters $k_{CA,0}*$ and $k_{CA,\infty}*$



depend on the molecular parameters and can be calculated theoretically or fitted experimentally (after the coupling between association and chemical activation has been accounted for). In practice one may try to represent the rate constants in the form of rate constants of separated processes $k_{Ass}^*$ and $k_{CA}^*$. Coupling these rate constants then leads to a full representation of the rate constants in terms of the six rate parameters $k_{Ass,0}$, $k_{Ass,\infty}$, $F_{Ass,c}$, $k_{CA,0}$, $k_{CA,\infty}$, and $F_{CA,c}$. If

one neglects the coupling and fits these parameters directly from the experiments (Miller and Klippenstein, 2001), however, one has to be aware of the fact that the values obtained do not correspond to those of separated, single-channel, association and chemical activation processes (for more details, see Troe, 2015).

As a consequence of the multistep character of complex-forming bimolecular reactions, a variety of temperature - and pressure – dependences of $k_{Ass}$ and $k_{CA}$ are observed. The low pressure limit of the total rate constants $k_{tot} = k_{Ass} + k_{CA}$, i.e., $k_{tot,0} = k_{CA,0} = k_1 k_5 / (k_{-1} + k_5)$, because of different energy – and angular momentum – dependences of the specific rate constants $k_1$, $k_{-1}$, and $k_5$, may increase or decrease with temperature, the latter with the possibility to a change

with an increase above a certain temperature. $k_{tot}$, as given above, may increase with pressure from $k_{CA,0}$ to $k_1$, with M = $H_2O$ often being a particularly efficient third body in the pressure – dependent range. The pressure dependence generally becomes less apparent with increasing temperature. Finally, the further fate of an addition product AB is of importance. It may be collisionally reactivated to energies where $k_5 >> k_{-1}$, such that formation of C + D is enhanced

(in comparison to energies where $k_5 << k_{-1}$). There is also the possibility that A-M  (or B-M) complexes are formed which react in a chaperon mechanism with B (or A) and then form products. M = $H_2O$ here again may be particularly efficient. Without detailed theoretical analysis, in general, it will be difficult to disentangle the intrinsic mechanism. Therefore, reference to theoretical work is given for selected reactions.


**2.5 Photochemical reactions**

The datasheets begin with a list of feasible primary photochemical transitions for wavelengths usually down to 170 nm, along with the corresponding enthalpy changes at 0 K where possible

or alternatively at 298 K. Calculated threshold wavelengths corresponding to these enthalpy changes are also listed, bearing in mind that the values calculated from the enthalpy changes at 298K are not true "threshold values". This is followed by tables which summarise the available experimental data for: (i) absorption cross sections and (ii) quantum yields. These data are



supplemented by a series of comments. The next table lists the preferred absorption cross

section data and the preferred quantum yields at appropriate wavelength intervals. For

absorption cross sections the intervals are usually 1 nm, 5 nm or 10 nm. Any temperature

dependence of the absorption cross sections is also given where possible. The aim in presenting

these preferred data is to provide a basis for calculating atmospheric photolysis rates. For

absorption continua the temperature dependence is often represented by Sulzer-Wieland type

expressions (Astholz et al., 1981). Alternately a simple empirical expression of the form:

$\log_{10}(\sigma_{T1}/\sigma_{T2}) = B*(T_1-T_2)$ is used. The comments again describe how the preferred data were

selected and include other relevant points. The photochemical datasheets are concluded with a

list of references.

**2.6 Conventions concerning absorption cross sections**

These are presented in the datasheets as "absorption cross sections per molecule, base e." They

are defined according to the equation:

$$I / I_0 = \exp(-\sigma[N]l) \qquad\qquad \text{Eq. (19)}$$

where $I_0$ and $I$ are the incident and transmitted light intensities, $[N]$ is the number concentration

of absorber (expressed in molecule $cm^{-3}$), $l$ is the path length (expressed in cm), and $\sigma$ is the

absorption cross section (units of $cm^2$ $molecule^{-1}$). Note that "molecule" is not a unit but is

included here for clarity. Other definitions and units are frequently quoted. The closely related

quantities "absorption coefficient" and "extinction coefficient" are often used, but care must be

taken to avoid confusion in their definition, see Calvert (1990) for definitions and discussion.

It is always necessary to know the units of concentration and of path length and the type of

logarithm (base e or base 10) corresponding to the definition.  The decadic molar absorption

coefficient, $\varepsilon$, is often quoted, particularly in the older literature, and is defined as:

$$\varepsilon = \{1/([A]l)\}\log_{10}(I_0/I), \qquad\qquad \text{Eq. (20)}$$

where $[A]$ is the concentration of the absorber expressed in units of moles per liter.  While $\varepsilon$ is

often called an extinction coefficient, the term "extinction" should more properly be used for

the sum of absorption and scattering. To convert from $\varepsilon$ (base 10, units of $dm^3$ $mol^{-1}$ $cm^{-1}$) to $\sigma$

(base e, units of $cm^2$ $molecule^{-1}$) multiply by $3.82\times10^{-20}$.





### 2.7 Assignment of uncertainties

Under the heading "reliability," estimates have been made of the absolute accuracies of the
preferred values of $k$ at 298 K and of the preferred values of E/R over the quoted temperature
range. The accuracy of the preferred rate coefficient at 298 K is quoted as the term $\Delta \log k$,
where $\Delta \log k = d$ and $d$ is defined by the equation, $\log k = c \pm d$. This is equivalent to the
statement that $k$ is uncertain to a factor of $f$, where $d = \log f$. The accuracy of the preferred value
of E/R is quoted as the term $\Delta(E/R)$, where $\Delta(E/R) = g$ and $g$ is defined by the equation E/R =
$h \pm g$. $d$ and $g$ are uncertainties corresponding approximately to 95% confidence limits. For
second-order rate coefficients listed in this evaluation, an estimate of the uncertainty at any
given temperature within the recommended temperature range may be obtained from the
equation:

$$\Delta \log k(T) = \Delta \log k(298 \text{ K}) + 0.4343 \{\Delta \text{ E/R}(1/T - 1/298 \text{ K})\} \qquad \text{Eq. (21)}$$

The assignment of these absolute uncertainties in $k$ and E/R is our subjective assessment. They
are not determined by a rigorous, statistical analysis of the database, which is generally too
limited to permit such an analysis. Rather, the uncertainties are based on our knowledge of the
techniques, the difficulties of the experimental measurements, the potential for systematic
errors, and the number of studies conducted and their agreement or lack thereof. Experience
shows that for rate measurements of atomic and free radical reactions in the gas phase, the
precision of the measurement, i.e. the reproducibility, is usually good. Thus, for single studies
of a particular reaction involving one technique, standard deviations, or even 95% confidence
limits, of $\pm 10\%$ or less are frequently reported in the literature. Unfortunately, when we compare
data for the same reaction studied by more than one group of investigators and involving
different techniques, the rate coefficients sometimes differ by a factor of 2 or even more. This
can only mean that one or more of the studies has involved large systematic uncertainty which
is difficult to detect. This is hardly surprising since, unlike molecular reactions, it is not always
possible to study atomic and free radical reactions in isolation, and consequently mechanistic
and other difficulties frequently arise. On the whole, our assessment of uncertainty limits tends
towards the cautious side. Thus, in the case where a rate coefficient has been measured by a
single investigation using one particular technique and is unconfirmed by independent work,
we typically suggest an uncertainty of a factor of 2.



In contrast to the usual situation for the rate coefficients of thermal reactions, where inter-comparison of results of a number of independent studies permits a realistic assessment of reliability, for many photochemical processes there is a scarcity of reliable data. Thus, we do not feel justified at present in assigning uncertainty limits to the parameters reported for the photochemical reactions.


**Author contribution**: All authors defined the scope of the work. AM, TJW, JNC, MEJ and RAC developed and drafted the data sheets and manuscript. All authors reviewed, refined, and revised the manuscript and data sheets.

**Competing interests:** The authors declare that they have no conflict of interest.

*Acknowledgements*. The Chairman and members of the Task Group wish to express their appreciation to IUPAC. for the financial help which facilitated the preparation of this evaluation. We also acknowledge financial support from the following organisations: EU Framework

Program 6 and 7 and Horizon 2020, the UK Natural Environmental Research Council; Swiss National Science Foundation, the Centre National de la Recherche Scientifique-Institut National des Sciences de l'Univers (CNRS-INSU), Orléans University and Observatoire des Sciences de l'Univers en région Centre (OSUC). We thank Cathy Boone and Phuong Ng for constructing, developing, and maintaining the website.






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

590





**Table 1. Summary of recommended rate coefficients for organic (≥C4) reactions**

*HO reactions based on datasheets in Supplement and on the IUPAC website updated in 2019*

| Reaction number | Reaction | $k_{298}$ cm$^3$ molecule$^{-1}$ s$^{-1}$ | $\Delta\log k_{298}$ | Temp. dependence of $k$/cm$^3$ molecule$^{-1}$ s$^{-1}$ | Temp. range/K | $\Delta(E/R)/K$ [a] |
|---|---|---|---|---|---|---|
| HOx_VOC7 | HO + CH$_3$CH$_2$CH$_2$CH$_3$ → H$_2$O + CH$_3$CH$_2$CH$_2$CH$_3$ → H$_2$O + CH$_3$CHCH$_2$CH$_3$ **overall** | $2.35 \times 10^{-12}$ | ± 0.06 | $9.8 \times 10^{-12}$ exp(-425/T) | 180-300 | ± 100 |
| HOx_VOC8 | HO + CH$_2$=C(CH$_3$)CH=CH$_2$ (isoprene) → products | $1.0 \times 10^{-10}$ | ± 0.06 | $2.1 \times 10^{-11}$ exp(465/T) | 240-630 | ± 150 |
| HOx_VOC9 | HO + α-pinene → products | $5.3 \times 10^{-11}$ | ± 0.08 | $1.3 \times 10^{-11}$ exp(410/T) | 240-360 | ± 100 |
| HOx_VOC14 | HO + CH$_3$CH$_2$CH$_2$CHO → products | $2.3 \times 10^{-11}$ | ± 0.08 | $5.8 \times 10^{-12}$ exp(410/T) | 250-430 | ± 250 |
| HOx_VOC15 | HO + CH$_2$=C(CH$_3$)CHO → products | $3.0 \times 10^{-11}$ | ± 0.08 | $8.4 \times 10^{-12}$ exp(380/T) | 230-380 | ± 100 |
| HOx_VOC20 | HO + CH$_3$C(O)CH$_2$CH$_3$ → products | $1.1 \times 10^{-12}$ | ± 0.10 | $1.5 \times 10^{-12}$ exp(-90/T) | 210-300 | ± 200 |
| HOx_VOC21 | HO + CH$_3$C(O)CH=CH$_2$ → products | $2.0 \times 10^{-11}$ | ± 0.10 | $2.6 \times 10^{-12}$ exp(610/T) | 230-380 | ± 200 |
| HOx_VOC22 | HO + pinonaldehyde → products | $3.9 \times 10^{-11}$ | ± 0.15 | | | |
| HOx_VOC27 | HO + CH$_3$CH$_2$CH$_2$CH$_2$OH → products | $8.5 \times 10^{-12}$ | ± 0.06 | $5.2 \times 10^{-12}$ exp(600/T) | 230-380 | ± 300 |
| HOx_VOC28 | HO + CH$_3$CH(OH)CH$_2$CH$_3$ → products | $8.7 \times 10^{-12}$ | ± 0.08 | $5.3 \times 10^{-12}$ exp(140/T) | 260-380 | ± 200 |
| HOx_VOC29 | HO + (CH$_3$)$_2$C(OH)CH=CH$_2$ → products | $6.3 \times 10^{-11}$ | ± 0.08 | $8.1 \times 10^{-12}$ exp(610/T) | 230-300 | ± 200 |
| HOx_VOC31 | HO + 3-methylfuran → products | $9.3 \times 10^{-11}$ | ± 0.15 | | | |
| HOx_VOC33 | HO + (CH$_3$)$_2$C(OH)CHO → products | $1.4 \times 10^{-11}$ | ± 0.10 | | | |
| HOx_VOC42 | HO + 1-C$_4$H$_9$ONO$_2$ → products | $1.6 \times 10^{-12}$ | ± 0.06 | | | |
| HOx_VOC43 | HO + 2-C$_4$H$_9$ONO$_2$ → products | $8.6 \times 10^{-13}$ | ± 0.15 | | | |
| HOx_VOC46 | HO + CH$_3$CH$_2$C(O)CH$_2$ONO$_2$ → products | $8.2 \times 10^{-13}$ | ± 0.30 | | | |
| HOx_VOC47 | HO + CH$_3$CH(ONO$_2$)C(O)CH$_3$ → products | $1.2 \times 10^{-12}$ | ± 0.30 | | | |
| HOx_VOC48 | HO + CH$_2$=C(CH$_3$)C(O)OONO$_2$ (MPAN) → products | $2.9 \times 10^{-11}$ | +0.2 -0.5 | | | |
| HOx_VOC60 | HO + (CH$_3$)$_3$CH → H$_2$O + (CH$_3$)$_3$C → H$_2$O + (CH$_3$)$_2$CHCH$_2$ **Overall** | $2.1 \times 10^{-12}$ | ± 0.04 | $5.4 \times 10^{-12}$ exp(-285/T) | 210-300 | ± 150 |
| HOx_VOC61 | HO + 2-methylpropene → products | $5.1 \times 10^{-11}$ | ± 0.04 | $9.4 \times 10^{-12}$ exp(505/T) | 290-430 | ± 200 |
| HOx_VOC62 | HO + 1-butene → products | $3.1 \times 10^{-11}$ | ± 0.06 | $6.6 \times 10^{-12}$ exp(465/T) | 290-430 | ± 150 |
| HOx_VOC63 | HO + cis-2-butene → products | $5.6 \times 10^{-11}$ | ± 0.10 | $1.1 \times 10^{-11}$ exp(485/T) | 290-430 | ± 200 |
| HOx_VOC64 | HO + trans-2-butene → products | $7.1 \times 10^{-11}$ | ± 0.06 | $1.1 \times 10^{-11}$ exp(553/T) | 290-430 | ± 200 |
| HOx_VOC66 | HO + CH$_3$C(O)C(O)CH$_3$ → products | $2.3 \times 10^{-13}$ | ± 0.06 | $5.25 \times 10^{-13}$ exp(-243/T) | 240-350 | ± 50 |
| HOx_VOC67 | HO + n-C$_3$H$_7$C(O)OH → products | $1.8 \times 10^{-12}$ | ± 0.15 | | | |
| HOx_VOC68 | HO + i-C$_3$H$_7$CHO → products | $2.6 \times 10^{-11}$ | ± 0.04 | $6.8 \times 10^{-12}$ exp(410/T) | 240-425 | ± 60 |
| HOx_VOC69 | HO + (CH$_3$)$_2$CHCH$_2$OH → products | $8.9 \times 10^{-12}$ | ± 0.08 | $2.73 \times 10^{-12}$ exp(352/T) | 240-370 | ± 120 |
| HOx_VOC70 | HO + (CH$_3$)$_3$COH → products | $1.1 \times 10^{-12}$ | ± 0.06 | $1.6 \times 10^{-12}$ exp(-121/T) | 240-314 | ± 75 |
| HOx_VOC76 | HO + C$_2$H$_5$CH(OH)CHO → C$_2$H$_5$CH(OH)CO + H$_2$O → C$_2$H$_5$C(OH)CHO + H$_2$O **overall** | $2.4 \times 10^{-11}$ | ± 0.10 | | | |
| HOx_VOC77 | HO + C$_2$H$_5$CH(OH)CH$_2$ONO$_2$ → products | $7.0 \times 10^{-12}$ | ± 0.20 | | | |
| HOx_VOC78 | HO + C$_2$H$_5$CH(ONO$_2$)CH$_2$OH → products | $7.4 \times 10^{-12}$ | ± 0.20 | | | |
| HOx_VOC79 | HO + CH$_3$C(O)CH(OH)CH$_3$ → H$_2$O + CH$_3$C(O)C(OH)CH$_3$ → products | | | | | |
| HOx_VOC84 | HO + α-terpinene → products **Overall** | $9.7 \times 10^{-12}$ $3.5 \times 10^{-10}$ | ± 0.10 ± 0.08 | $1.24 \times 10^{-12}$ exp(612/T) | 280-350 | ± 350 |



| ID | Reaction | $k$ | $\pm$ | $k(T)$ | $T$ range | $\pm$ |
|---|---|---|---|---|---|---|
| HOx_VOC85 | HO + $\gamma$-terpinene $\rightarrow$ products | $1.7 \times 10^{-10}$ | $\pm 0.10$ | | | |
| HOx_VOC86 | HO + terpinolene $\rightarrow$ products | $2.2 \times 10^{-10}$ | $\pm 0.15$ | | | |
| HOx_VOC87 | HO + $\alpha$-phellandrene $\rightarrow$ products | $3.2 \times 10^{-10}$ | $\pm 0.08$ | | | |
| HOx_VOC88 | HO + $\beta$-phellandrene $\rightarrow$ products | $1.7 \times 10^{-10}$ | $\pm 0.15$ | | | |
| HOx_VOC89 | HO + $\alpha$-cedrene $\rightarrow$ products | $6.7 \times 10^{-11}$ | $\pm 0.10$ | | | |
| HOx_VOC90 | HO + longifolene $\rightarrow$ products | $4.7 \times 10^{-11}$ | $\pm 0.10$ | | | |
| HOx_VOC91 | HO + $\alpha$-copaene $\rightarrow$ products | $9.0 \times 10^{-11}$ | $\pm 0.10$ | | | |
| HOx_VOC92 | HO + $\beta$-caryophyllene $\rightarrow$ products | $2.0 \times 10^{-10}$ | $\pm 0.15$ | | | |
| HOx_VOC93 | HO + $\alpha$-humulene $\rightarrow$ products | $2.9 \times 10^{-10}$ | $\pm 0.10$ | | | |
| HOx_VOC99 | HO + $\beta$-pinene $\rightarrow$ products | $7.6 \times 10^{-11}$ | $\pm 0.05$ | $1.62 \times 10^{-11}\exp(460/T)$ | 240-420 | $\pm 150$ |
| HOx_VOC100 | HO + limonene $\rightarrow$ products | $1.65 \times 10^{-10}$ | $\pm 0.05$ | $3.41 \times 10^{-11}\exp(470/T)$ | 220-360 | $\pm 150$ |
| HOx_VOC101 | HO + camphene $\rightarrow$ products | $5.2 \times 10^{-11}$ | $\pm 0.10$ | $4.14 \times 10^{-12}\exp(754/T)$ | 280-320 | $\pm 100$ |
| HOx_VOC102 | HO + 2-carene $\rightarrow$ products | $8.0 \times 10^{-11}$ | $\pm 0.15$ | | | |
| HOx_VOC103 | HO + 3-carene $\rightarrow$ products | $8.3 \times 10^{-11}$ | $\pm 0.06$ | $2.5 \times 10^{-11}\exp(357/T)$ | 230-360 | $\pm 50$ |
| HOx_VOC104 | HO + $\beta$-myrcene $\rightarrow$ products | $2.1 \times 10^{-10}$ | $\pm 0.15$ | | | |
| HOx_VOC105 | HO + $\beta$-ocimene $\rightarrow$ products | $2.8 \times 10^{-10}$ | $\pm 0.15$ | $4.0 \times 10^{-11}\exp(579/T)$ | 310-430 | $\pm 150$ |
| HOx_VOC106 | HO + $\beta$-sabinene $\rightarrow$ products | $1.2 \times 10^{-10}$ | $\pm 0.15$ | | | |
| HOx_VOC107 | HO + $\alpha$-farnesene $\rightarrow$ products | $2.2 \times 10^{-10}$ | $\pm 0.30$ | $2.2 \times 10^{-10}$ | 298-430 | $\pm 200$ |
| HOx_VOC108 | HO + $\beta$-farnesene $\rightarrow$ products | $2.3 \times 10^{-10}$ | $\pm 0.30$ | $2.3 \times 10^{-10}$ | 298-430 | $\pm 200$ |
| HOx_VOC109 | HO + $\alpha$-terpineol $\rightarrow$ products | $1.9 \times 10^{-10}$ | $\pm 0.30$ | | | |
| HOx_AROM1 | HO + $C_6H_6$ (benzene) $\rightarrow H_2O + C_6H_5$ $\rightarrow HOC_6H_6$ | $8 \times 10^{-15}$ | $\pm 0.50$ | $3.8 \times 10^{-11}\exp(-2520/T)$ | 330-1410 | $\pm 300$ |
| Overall | | $1.2 \times 10^{-12}$ | $\pm 0.06$ | $2.3 \times 10^{-12}\exp(-190/T)$ | 230-350 | $\pm 200$ |
| HOx_AROM2 | HO + $C_6H_5CH_3$ (toluene) $\rightarrow H_2O + C_6H_5CH_2$ $\rightarrow HOC_6H_5CH_3$ | $3.5 \times 10^{-13}$ | $\pm 0.20$ | $2.5 \times 10^{-11}\exp(-1270/T)$ | 310-1050 | $\pm 200$ |
| Overall | | $5.6 \times 10^{-12}$ | $\pm 0.06$ | $1.8 \times 10^{-12}\exp(340/T)$ | 210-350 | $\pm 200$ |
| HOx_AROM3 | HO + $m$-$CH_3C_6H_4OH$ ($m$-cresol) $\rightarrow H_2O + CH_3C_6H_4O$ $\rightarrow H_2O + CH_2C_6H_4OH$ $\rightarrow CH_3C_6H_4(OH)_2$ | | | | | |
| Overall | | $6.2 \times 10^{-11}$ | $\pm 0.10$ | $2.4 \times 10^{-12}\exp(965/T)$ | 290-350 | $\pm 600$ |
| HOx_AROM4 | HO + $o$-$CH_3C_6H_4OH$ ($o$-cresol) $\rightarrow H_2O + CH_3C_6H_4O$ $\rightarrow H_2O + CH_2C_6H_4OH$ $\rightarrow CH_3C_6H_4(OH)_2$ | | | | | |
| Overall | | $4.2 \times 10^{-11}$ | $\pm 0.10$ | $1.6 \times 10^{-12}\exp(970/T)$ | 290-350 | $\pm 600$ |
| HOx_AROM5 | HO + $p$-$CH_3C_6H_4OH$ ($p$-cresol) $\rightarrow H_2O + CH_3C_6H_4O$ $\rightarrow H_2O + CH_2C_6H_4OH$ $\rightarrow CH_3C_6H_4(OH)_2$ | | | | | |
| Overall | | $4.8 \times 10^{-11}$ | $\pm 0.10$ | $1.9 \times 10^{-12}\exp(970/T)$ | 290-350 | $\pm 600$ |
| HOx_AROM6 | HO + $C_6H_5OH$ (phenol) $\rightarrow H_2O + C_6H_5O$ $\rightarrow H_2O + C_6H_4OH$ $\rightarrow HOC_6H_5OH$ | | | | | |
| Overall | | $2.8 \times 10^{-11}$ | $\pm 0.08$ | $4.7 \times 10^{-13}\exp(1220/T)$ | 290-350 | $\pm 600$ |
| HOx_AROM7 | HO + 1,2-dihydroxybenzene (1,2-$C_6H_4(OH)_2$) $\rightarrow$ products | $1.0 \times 10^{-10}$ | $\pm 0.15$ | | | |
| HOx_AROM8 | HO + 1,2-dihydroxy-3-methylbenzene $\rightarrow$ products | $2.0 \times 10^{-10}$ | $\pm 0.15$ | | | |
| HOx_AROM9 | HO + 1,2-dihydroxy-4-methylbenzene $\rightarrow$ products | $1.5 \times 10^{-10}$ | $\pm 0.15$ | | | |





Atmospheric Chemistry and Physics Discussions — Open Access / EGU

| Label | Reaction | $k$ | Unc. | Expression | $T$ range | Unc. |
|---|---|---|---|---|---|---|
| HOx_AROM10 | HO + 3-methyl-2-nitrophenol → products | $3.4 \times 10^{-12}$ | ± 0.30 | | | |
| HOx_AROM11 | HO + 4-methyl-2-nitrophenol → products | $3.4 \times 10^{-12}$ | ± 0.15 | | | |
| HOx_AROM12 | HO + 5-methyl-2-nitrophenol → products | $6.4 \times 10^{-12}$ | ± 0.20 | | | |
| HOx_AROM13 | HO + 6-methyl-2-nitrophenol → products | $2.5 \times 10^{-12}$ | ± 0.30 | | | |
| HOx_AROM14 | HO + 1,4-benzoquinone → products | $4.6 \times 10^{-12}$ | ± 0.15 | | | |
| HOx_AROM15 | HO + methyl-1,4-benzoquinone → products | $2.3 \times 10^{-11}$ | ± 0.15 | | | |
| HOx_AROM16 | HO + $C_6H_5NO_2$ (nitrobenzene) → products | $1.4 \times 10^{-13}$ | ± 0.20 | $6.0 \times 10^{-13} \exp(-440/T)$ | 250-370 | ± 300 |
| HOx_AROM17 | HO + 3-nitrotoluene → products | $1.2 \times 10^{-12}$ | ± 0.50 | | | |
| HOx_AROM18 | HO + cis-CHOCH=CHCHO → products | $5.7 \times 10^{-11}$ | ± 0.20 | | | |
| HOx_AROM19 | HO + trans-CHOCH=CHCHO → products | $\geq 2 \times 10^{-11}$ | | | | |
| HOx_AROM20 | HO + 3H-furan-2-one → products | $4.8 \times 10^{-11}$ | ± 0.20 | | | |
| HOx_AROM21 | HO + furan-2,5-dione → products | $1.4 \times 10^{-12}$ | ± 0.20 | | | |
| HOx_AROM22 | HO + $CH_3C(O)CH=CHCHO$ (cis/trans-4-oxopent-2-enal) → products | $6.2 \times 10^{-11}$ | ± 0.20 | | | |
| HOx_AROM25 | HO + $C_6H_5CHO$ (benzaldehyde) → products | $1.26 \times 10^{-11}$ | ± 0.08 | $5.9 \times 10^{-12} \exp(225/T)$ | 290-350 | ± 170 |
| HOx_AROM26 | HO + $C_6H_5CH_2OH$ (benzyl alcohol) → products | $2.7 \times 10^{-11}$ | ± 0.08 | | | |

**NO3 reactions based on datasheets in Supplement and on the IUPAC website updated in 2019**

| Label | Reaction | $k$ | Unc. | Expression | $T$ range | Unc. |
|---|---|---|---|---|---|---|
| NO3_VOC26 | NO3 + 2-methylpropane, $(CH_3)_3CH$ → products | $1.1 \times 10^{-16}$ | ± 0.10 | | | |
| NO3_VOC27 | NO3 + 2-methylpropene $((CH_3)_2C=CH_2)$ → products | $3.4 \times 10^{-13}$ | ± 0.10 | | | |
| NO3_VOC28 | NO3 + 1-butene → products | $1.3 \times 10^{-14}$ | ± 0.10 | $3.0 \times 10^{-12} \exp(-3050/T)$ | 290-430 | ± 300 |
| NO3_VOC29 | NO3 + cis-2-butene → products | $3.5 \times 10^{-13}$ | ± 0.10 | $3.2 \times 10^{-13} \exp(-950/T)$ | 230-480 | ± 200 |
| NO3_VOC30 | NO3 + trans-2-butene → products | $3.9 \times 10^{-13}$ | ± 0.08 | $\{1.78\times10^{-12}\exp(-530/T) + 1.28\times10^{-14}\exp(570/T)\}$ | 200-380 | |
| NO3_VOC33 | NO3 + d-limonene → products | $1.2 \times 10^{-11}$ | ± 0.12 | | | |
| NO3_VOC34 | NO3 + 2-carene → products | $2.0 \times 10^{-11}$ | ± 0.12 | | | |
| NO3_VOC35 | NO3 + 3-carene → products | $9.1 \times 10^{-12}$ | ± 0.12 | | | |
| NO3_VOC36 | NO3 + β-pinene → products | $2.5 \times 10^{-12}$ | ± 0.12 | | | |
| NO3_VOC37 | NO3 + myrcene → products | $1.1 \times 10^{-11}$ | ± 0.12 | | | |
| NO3_VOC38 | NO3 + sabinene → products | $1.0 \times 10^{-11}$ | ± 0.10 | | | |
| NO3_VOC39 | NO3 + ocimene, cis and trans → products | $2.2 \times 10^{-11}$ | ± 0.15 | | | |
| NO3_VOC40 | NO3 + α-terpinene → products | $1.8 \times 10^{-10}$ | ± 0.25 | | | |
| NO3_VOC41 | NO3 + γ-terpinene → products | $2.9 \times 10^{-11}$ | ± 0.12 | | | |
| NO3_VOC42 | NO3 + α-phellandrene → products | $7.3 \times 10^{-11}$ | ± 0.15 | | | |
| NO3_VOC43 | NO3 + terpinolene → products | $9.7 \times 10^{-11}$ | ± 0.25 | | | |
| NO3_VOC46 | NO3 + camphene → products | $6.6 \times 10^{-13}$ | ± 0.10 | | | |
| NO3_VOC47 | NO3 + β-caryophyllene → products | $1.9 \times 10^{-11}$ | ± 0.25 | | | |
| NO3_VOC48 | NO3 + α-cedrene → products | $8.2 \times 10^{-12}$ | ± 0.25 | | | |
| NO3_VOC49 | NO3 + α-humulene → products | $3.5 \times 10^{-11}$ | ± 0.25 | | | |
| NO3_VOC50 | NO3 + α-copaene → products | $1.6 \times 10^{-11}$ | ± 0.25 | | | |
| NO3_VOC51 | NO3 + longifolene → products | $6.8 \times 10^{-13}$ | ± 0.25 | | | |
| NO3_VOC52 | NO3 + isolongifolene → products | $3.9 \times 10^{-12}$ | ± 0.25 | | | |
| NO3_VOC53 | NO3 + alloisolongifolene → products | $1.4 \times 10^{-12}$ | ± 0.25 | | | |
| NO3_VOC54 | NO3 + α-neoclovene → products | $8.25 \times 10^{-12}$ | ± 0.25 | | | |



| | | | |
|---|---|---|---|
| NO3_VOC55 | NO$_3$ + valencene $\rightarrow$ products | $7.9 \times 10^{-12}$ | $\pm 0.25$ |
| NO3_VOC56 | NO$_3$ + $\alpha$-terpineol $\rightarrow$ products | $1.7 \times 10^{-11}$ | $\pm 0.25$ |
| NO3_AROM1 | NO$_3$ + C$_6$H$_6$ (benzene) $\rightarrow$ products | $< 3 \times 10^{-17}$ | |
| NO3_AROM2 | NO$_3$ + C$_6$H$_5$CH$_3$ (toluene) $\rightarrow$ products | $7.8 \times 10^{-17}$ | $\pm 0.25$ |
| NO3_AROM3 | NO$_3$ + $m$-CH$_3$C$_6$H$_4$OH ($m$-cresol) $\rightarrow$ CH$_3$C$_6$H$_4$O + HNO$_3$ $\rightarrow$ other products | | |
| | **Overall** | $1.0 \times 10^{-11}$ | $\pm 0.15$ |
| NO3_AROM4 | NO$_3$ + $o$-CH$_3$C$_6$H$_4$OH ($o$-cresol) $\rightarrow$ CH$_3$C$_6$H$_4$O + HNO$_3$ $\rightarrow$ other products | | |
| | **Overall** | $1.4 \times 10^{-11}$ | $\pm 0.15$ |
| NO3_AROM5 | NO$_3$ + $p$-CH$_3$C$_6$H$_4$OH ($p$-cresol) $\rightarrow$ CH$_3$C$_6$H$_4$O + HNO$_3$ $\rightarrow$ other products | | |
| | **Overall** | $1.1 \times 10^{-11}$ | $\pm 0.15$ |
| NO3_AROM6 | NO$_3$ + C$_6$H$_5$OH (phenol) $\rightarrow$ C$_6$H$_5$O + HNO$_3$ $\rightarrow$ other products | | |
| | **Overall** | $3.8 \times 10^{-12}$ | $\pm 0.15$ |
| NO3_AROM7 | NO$_3$ + 1,2-dihydroxybenzene (1,2-C$_6$H$_4$(OH)$_2$) $\rightarrow$ products | $9.9 \times 10^{-11}$ | $\pm 0.15$ |
| NO3_AROM8 | NO$_3$ + 1,2-dihydroxy-3-methylbenzene $\rightarrow$ products | $1.7 \times 10^{-10}$ | $\pm 0.15$ |
| NO3_AROM9 | NO$_3$ + 1,2-dihydroxy-4-methylbenzene $\rightarrow$ products | $1.5 \times 10^{-10}$ | $\pm 0.15$ |

*Photochemical reactions based on datasheets in Supplement and on the IUPAC website updated in 2019*

| | |
|---|---|
| P23 | CH$_3$C(O)C(O)CH$_3$ + h$\nu$ $\rightarrow$ products |
| P24 | i-C$_3$H$_7$CHO + h$\nu$ $\rightarrow$ products |
| P26 | cis/trans-but-2-enedial + h$\nu$ $\rightarrow$ products |
| P27 | 4-oxopent-2-enedial + h$\nu$ $\rightarrow$ products |
| P28 | 2-nitrophenol + h$\nu$ $\rightarrow$ products |
| P30 | benzaldehyde + h$\nu$ $\rightarrow$ products |
| P31 | 3-methyl-2-nitrophenol + h$\nu$ $\rightarrow$ products |
| P32 | 4-methyl-2-nitrophenol + h$\nu$ $\rightarrow$ products |

[a] The cited uncertainty corresponds approximately to 95% confidence limits.