# Peer review of "Evaluated kinetic and photochemical data for atmospheric chemistry: Volume VIII - gas phase reactions of organic species with four, or more, carbon atoms ( $\geq C_4$ )"

_Atmospheric Chemistry and Physics, 2020_

## Referee Comment (RC1) · Anonymous Referee #1 · 25 Sep 2020

This IUPAC evaluation (the eighth in the series) is clearly publishable in ACP. The evaluation panel provides an invaluable service to the atmospheric chemistry community, and the publication of their recommendations in ACP is an entirely appropriate mechanism for broad dissemination of this information. The text is good, and the summary recommendations all seemed correct. I have only one suggestion for improvement. It is obvious that the panel cannot review all C4+ species for which data exist, but can the authors provide a summary in the text of the scope of the species evaluated? Or the rationale for determining which species are included in the review?

---

## Referee Comment (RC2) · Anonymous Referee #2 · 19 Oct 2020

As stated in the abstract, this is the eight overview article published in ACP on the detailed evaluation work carried out by the IUPAC Task Group on Atmospheric Chemical Kinetic Data Evaluation, focusing on the gas phase reactions of organic species with four, or more, carbon atoms. The work carried out by this group is of critical importance to the atmospheric chemistry community, where these evaluations drive our current understanding of tropospheric and stratospheric chemistry represented in the chemical mechanisms that underpin all air quality and climate science and policy models.

This overview article summarises the datasheets and recommendations on the reac-

tions of selected tropospherically important $\geq$ C4 closed-shell organic species with HO and NO3 radicals, and their photolysis. The datasheets and recommendations discussed are available, along with recommendations of other atmospherically important gas, heterogeneous and aqueous phase reactions on a dedicated webpage, which is updated periodically. The article itself is a well written and clear summary which certainly should be published in ACP. One suggestion is that it would be useful to the reader if an additional table is added which summarises the range of VOCs covered, e.g. how many alkanes, alkenes, aromatics, oxygenated species, organic nitrates and nitro compounds, etc. . . , with the supplementary datasheets logically ordered as appropriate.

---

## Referee Comment (RC3) · Anonymous Referee #3 · 20 Oct 2020

This paper presents the eighth evaluation of kinetic and photochemical data by the IUPAC Task Group on Atmospheric Chemical Kinetic Data Evaluation. The work performed by this evaluation panel is of prime importance for the scientific community of atmospheric sciences, in particular for the development of chemical schemes in atmospheric models but also for lab and field experimental studies. Recommendations are well justified in the supplement document and appear reasonable. This paper, which complements well the online database, is very useful for users as it includes a theoretical section on the kinetic treatment of the various types of atmospheric

reactions in the gas phase. It also includes a summary of recommended kinetic parameters as well as detailed information upon which these recommendations are made. This paper is clear and well written. For rate constants that have been subject to re-evaluation, I would suggest to provide in the paper, or in the interactive database, the former value (for example in brackets). This could be very useful for the users as it would improve the traceability of the changes made. As a recommendation, this is not a critical point for the publication of the paper which can be published in its current state. However, I think that it could be useful for the users. In conclusion, I fully support the publication of this paper.

Please also note the supplement to this comment:
https://acp.copernicus.org/preprints/acp-2020-940/acp-2020-940-RC3-supplement.pdf
* * *

---

## Referee Comment (RC4) · Cornelius Zetzsch (Referee) · 22 Oct 2020

Referees Cornelius Zetzsch and Geert Moortgat

IUPAC data evaluation. Review of Manuscript acp-2020-940

This current article is a supplement to the "Evaluated kinetic and photochemical data for atmospheric chemistry, Volume II; gas-phase reactions of organic species" (Atkinson *et al.,* 2006). It updates existing data sheets, presents several new ones and continues to be a valuable tool for the scientific community. The kinetic and photochemical datasheets of the present work are (or are about to be) accessible in the internet (www.iupac.pole-ether.fr). The manuscript is clearly publishable in ACP, where the readers may take the opportunity to add hints at errors and other work for consideration (further to corresponding with the lead author or any other member of the task group).

The article starts with an introduction into the series with volumes on

$O_x$, $HO_x$, $NO_x$, and $SO_x$ species

organic species (the topic of the present work)

inorganic halogens

organic halogen species

heterogeneous reactions on solid substrates

heterogeneous reactions with liquid substrates

and, in preparation, reactions of organic species with ozone and chemistry of Criegee intermediates

It continues to present a guide to the datasheets with precise information on the equations of the functions employed and to the assignment of uncertainties in the recommendations of rate coefficients and Arrhenius parameters for room temperature and below.

Moreover, the supplement shows Arrhenius diagrams for more than 40 molecules (how many alkanes, olefins and terpenoids, alcohols, aldehydes, acids, nitro compounds, aromatics?) and diagrams of UV absorption cross-sections for 8 molecules. You may wish to list the Arrhenius diagrams with trivial names of the molecules on page 2, line 69 and might point out features of positive and negative temperature dependence (and convex and concave curvature), to be observed in the supplement.

General and specific comments and technical corrections:

P. 1, L. 25: The abstract might mention that the present work is a continuation of Volume II (Atkinson et al., 2006) with new (which molecules?) and updated data sheets for Appendix A2 (HO reactions), A3 ($NO_3$ reactions) and A8 (photochemical processes).

P.1, l. 26: Have all of the data sheets of the present supplement been evaluated in 2019 or should the date be given individually for each data sheet or should the IUPAC website be consulted?

P. 3, line 73:  gas-phase might be cancelled (see title)

P. 4: There are further abbreviations in the supplement, which could be listed, such as API, FEP, CEAS, IBBCEAS, ToF, EUPHORE

P. 9, l. 285: Are there appropriate, obvious examples for combination reactions without a barrier?

P. 12, l. 386: In which cases is $H_2O$ a particularly efficient third body?

P.12, l. 398: Photochemical *transitions* or *processes*? The data sheets comprise cases, where available UV spectra have not been discussed down to 170 nm, where UV spectra for, e.g., 2-nitrophenol are available (http://satellite.mpic.de/spectral_atlas).

P. 14, l. 460: "Unfortunately" or better: "On the other hand" or "Moreover" (?). This situation stimulates further investigation with improved techniques, wider ranges of concentration, total pressure and temperature, or detailed consideration of the tail of non-exponential decays as biexponential, triexponential, multiexponential or mixed with second order components

P. 14, l. 463-466: Some more reasons for such differences of more than a factor of two could be listed here, such as insufficient purity of compounds, unknown/unexpected impurities, wall adsorption and decomposition, insufficient time resolution for the initial elementary step, opposing and parallel reactions or unexpected decomposition products from the primary product

P. 15, line 473:  "scarcity of reliable data"  =>  could you delete "reliable" (?)

P. 15, line 483:  IUPAC. For the  => IUPAC for the

P. 16. line 524: "J. Phys. Chem. Ref. Data" occurs twice

P. 17, line 550: One might expect the series of evaluations by Atkinson (monographs) and by Calvert et al., most of which are mentioned in the supplement in several datasheets, to be cited here as well.

P. 19-22: Overall or overall?

P. 22: Table 1 should end with the $NO_3$ reactions and the footnote, and the photochemical reactions should be a separate Table 2 (with different entries, such as absorption cross-sections and quantum yields?).

**Supplement**

**General remarks on the text**

It occurs in several places of the supplement (unlike the main manuscript) that the list of authors in the references does not end with a colon before the journal follows (in the photolysis section mainly).

Exponents and their numerals should not appear in separate lines.

The dates of final evaluation (and recommendation) should be mentioned in the heading of every data sheet, like those available on the website.

Unlisted authors (*et al.*), isomers *cis*, *trans, H-, n-* and *i-* should be marked by italic fonts in the whole text, in the tables and in the comments.

The date of the IUPAC recommendations (IUPAC and year instead of Arrhenius fit (or biexponential fit on page 212) should appear in the figures

The use of upper and lower case letters in headings, footnotes and figure captions is inconsistent.

Pages 1-3  Some simple structures of aliphatics are missing, and trivial names of several molecules could be added as explanation in brackets, even for catechol.

Tables on Pages 5 (*n*-butane), 141 (benzene), 147 (toluene): Recent work exists on relative rate constants for 17 hydrocarbons, including *n*-butane, benzene and toluene. These have been investigated at 248 and 288 K in a smog chamber by Han et al. (L. Han, F. Siekmann, C. Zetzsch: Atmosphere 9, 320, 2018, doi:10.3390/atmos9080320) in reasonable agreement with the IUPAC recommendations.

Page 141, Line 4110:    => pressure. The non-exponential decays were shown to be biexponential by Wahner and Zetzsch (1983), Knispel et al. (1990), and Bohn

and Zetzsch (1999), according to the analytical solution of the differential equation system, where the back-decomposition of the HO-benzene adduct, leads to an equilibration of OH with the adduct, considering the abstraction channel as irreversible loss process of OH and the adduct.

**Figures**

Unlisted authors (*et al.*, sometimes those who spent their time performing the experiments or improving the equipment) are missing in several figures (especially the $NO_3$ section), in some instances the second author as well.

Not in all cases are the rate coefficients and Arrhenius expressions derived by other authors listed in the tables and shown in the graphs (this would help the reader even if no IUPAC recommendation is made for a single study of the temperature dependence).

This holds even more so if an expression derived by other authors is adopted by IUPAC (e.g. *n*-butane, adopted from Donahue and Clarke, 2004).

P. 8 (*n*-Butane): correct Clark into Clarke, add tick marks or gridlines of the logarithmic scale. Gridlines might be added to all figures

P. 17 (Isoprene) The reaction HO + Isoprene does not appear in the figure as a label. In the same fashion, labels with the reactions are missing on pages 22, 28, 79, 118, 122, 125, 128, 130, 132, 215, 217, 224, 226, 231, 233, 235, 237.

Delete black boxes around the reaction labels on pages 83, 86 and 197

Except for a few aromatics, the abstraction channel has not been shown in the Arrhenius diagrams. Equilibration of OH and $NO_3$ with the adduct may cause the negative activation energy for olefinic compounds and a concave curvature at high temperatures. Have available high-temperature data or non-exponential decays at intermediate temperatures been considered  in all cases?

**Technical corrections**

Line 2853:  at al.  => *et al.*

Line 2969:  at al.  => *et al.*

Line 3235: at al.  => *et al.*

Lines 3450 and 6418: The asterisk might be explained by a footnote, or one might ask the authors for the correct temperature

Line 4095:  non-exponential  => biexponential

Line 4117: non-exponential  => biexponential

 Line 4194: non-exponential  => biexponential

Line 4290: non-exponential  => biexponential

Line 4405:  non-exponential  => biexponential

Line 4211:  Somerlade => Sommerlade

Line 4789:  non-exponential  => biexponential

Line 5656:  to appear   => add year of appearance (2011)

Line 5834:  => over 2-methylpropene

Line 5838: => over 2-methylpropene

Line 5840: => of 2-methylpropene

Line 6153:  => Bunsenges.

Lines 6186, 6286, 6525, 6600, 6763, 6836, 6968: $NO_3$. So that….  =>  $NO_3$, so that

Line 6436: 1 bar

Lines 6542, 6781, 6861, 6922:   accurate  =>  absolute

Line 6746: Bunsen-Ges. => Bunsenges.

Line 6876:  Bunsen-Ges. => Bunsenges.

Line 6879:  cancel VOC55

Line 7030:  8-methylene..

Line 7254:  7-tetra

Line 7290: a-dimethyl

Line 7329:  a-hexa..

Line 7463:  was obtained.

Line 7533:  agreement, and  => agreement and  (?)

Line 7566:  RRGC  => RR-GC

Line 7617:   butene.

Line 7622:   5-methyl…

Line 7679:  butene.

**Photochemistry section**

**General comments**

**1 Selection criteria**

In this supplement of Volume II (Atkinson *et al.*, 2006), 8 organic species with four, or more, carbon atoms ($\geq C_4$) were evaluated.

What are the criteria of the selection of those eight organic species? More photolysis studies of organic species ($\geq C_4$) have been published in the literature, which could have been evaluated in this supplement.

Examples are:  *n*-butanal and *n*-pentanal (Tadic *et al.*, 2001a), *n*-hexanal (Tadic *et al.*, 2001b), *n*-heptanal (Tadic *et al.*, 2002), *n*-octanal (Tadic *et al.*, 2011), *trans*-crotonaldehyde (Magneron *et al.*, 2002) and methyl ethyl ketone (Nádasdi *et al.*, 2010)

References

Magneron, I., Thévenet, R., Mellouki, A., Le Bras, G., Moortgat, G. K. and Wirtz, K.: J. Phys., Chem., A, 106, 2526 (2002).

Nádasdi, R., Zügner, G. L., Farkas, M., Dóbé, S., Maeda, S. and Morokuma, K.: Chem. Phys. Chem., 11, 3883 (2010).

Tadic, J., Juranic, I. and Moortgat, G. K.: J. Photochem. Photobiol. A: Chem., 143, 169 (2001a).

Tadic, J. M., Juranic, I. and Moortgat, G. K.: Molecules, 6, 287 (2001b).

Tadic, J. M., Juranic, I. O. and Moortgat, G. K.: J. Chem. Soc., Perkin Trans., 2, 135 (2002).

Tadic, J. M., Lai Xu, Houk, K. N. and Moortgat, G. K.: J. Org. Chem., 76, 1614 (2011).

**2  Presentation**

a) In the Summary page (p 271, line 8010) one product channel is given, which is not correct. One ought to replace this by "products", as is shown in the datasheets, or add the other product channels

b) The head texts of the current datasheets are different from those presented in Volume II (Atkinson *et al.*, 2006). The text that appeared on the website   (iupac.pole-ether.fr) contains additional information, such as the update date

c) In the title molecule, it would be advisable to add the trivial name

d) The presentation section starts with "Primary photochemical transitions" However, in older datasheets of Volume IV (Atkinson *et al.*, 2008), this title was named "Primary photochemical processes". Would this title be more appropriate?

e) In all photochemical datasheets, the substances appear above the figure

**3   Technical corrections**

A) Throughout the text in Volumes II and VIII, **the references** are not presented uniformly. It is advisable to use the same citation style throughout the manuscript, including the supplement

**B) Units.**

The text should use for the cross-section units

   $cm^2$ molecule$^{-1}$   and not   $cm^2$molecule$^{-1}$   nor   $cm^2$/molecule

Additional figures of the spectra on a logarithmic scale (or a reference to the Spectral Atlas, where these are an option) might be useful.

Term symbols of the transitions (absorption bands displayed in the figures) and rough estimates of the oscillator strengths would be useful and a key to the photolytic processes.

**C) Individual remarks**

**P23**

Line 8022    add trivial name: biacetyl

Line 8027    align reaction products

Line 8063    correct $p = \infty$    into    $p \rightarrow \infty$

Line 8071    remove one comma: Horowitz *et al.* (2001), which are….

Line 8092    correct: Barnes, I. and Becker K. H.,

Line 8094    correct: Calvert, J. G. and Pitts Jr., J. N.,

Line 8098    correct: Ravishankara, A. R. and Burkholder, J. B.,

Line 8104    correct: Absorption spectrum of biacetyl

**P24**

Line 8111    add trivial name: *i*-butyraldehyde

Line 8114    align reaction products

Line 8130    correct: with a resolution

Line 8135    remove comma: … determined from measurements…

Line 8153/54    rewrite: …except for very slightly at 330.5 nm . >>>>

     ….except at 330.5 nm, where a minimal pressure dependence was observed.

Line 8174    replace:… better than 4 %... by … smaller than 4%

Line 8197    correct: …Francisco, J. S.: J. Phys. Chem. A, …. 2002.

Line 8200    correct: Calvert, J. G. and Pitts Jr., J. N.,

Line 8206    correct: Absorption spectrum of *i*-butyraldehyde

**P26**

Line 8213    add trivial name.

     Note: two different names appear in the text:

         butenendial and butene-2-dial

     It is assumed that butene-2-dial is correct, and should be corrected throughout the text.

Line 8213    move arrow, and align product channel numbers

Line 8219    correct reference:  Hufford *et al.*, 1952.

Line 8222    the transitions "*cis-/trans & trans/-cis*" should be in italics font

Line 8226    the Comments a, b, c, d, and e, are erroneously labeled a, b, c, c, and d

Lines 8230, 8232, 8240   correct units to:  $cm^2$ $molecule^{-1}$

Line 8232    move right bracket: purified (crystalline) fumaric dialdehyde

Lines 8232 and 8239      correct: cross-sections (not cross sections)

Line 8242    insert commas: at 193 nm, HCO produced by the Cl + HCHO reaction, was…

Line 8243    delete " in"

Line 8247    correct: "was" into "were"

Line 8253    correct:  assigned

Lines 8255 and 8282: add year (1994) of reference

Line 8257    change temperature to *298* K

Line 8271    correct  Fig 1   to   Fig. 1

Line 8278    insert comma after …limits >>>… limits,

Line 8287    correct:  ….  Barnes, I., Becker, K. H. and Wiesen, P., Environ. Sci. Technol.

Line 8289    correct:  …Chem. Phys. Lett.

Line 8242    enter space between first names of authors

Line 8305    correct:  Absorption spectrum…

**P27**

Line 8311    Note: three different names appear in the text:

4-oxo-pent-2-enal , 4-oxo-2-pentenal and

4-oxo-penten-2-dial (see figure caption)

Which is correct?, This should be corrected throughout the text.

Line 8313    align product channels

Line 8322    the transitions "*cis-trans & trans-cis*" should be *in italics*

Line 8328 and further throughout this comment section:
correct cross-section (not cross section)

Lines 8352 and 8354    correct:    5-methyl-3*H*-furan-2-one,    not

5-Methyl-3H-furan-2-one

Line 8358    add year (1994) of reference

Line 8370    change temperature to *298* K

Line 8378    correct:    … study of *the* gas-phase…

Line 8383    enter space:  193 nm

Line 8395    insert comma after:  However,…

Line 8397    correct Bierbach et al. (1994)

Line 8401    add year (1994) of reference

Line 8411    correct figure caption: .. spectrum of  [enter correct name]

**P28**

Line 8421:

Absorption cross-section data and quantum yields of M. Sangwan and L. Zhu ("Absorption cross sections of 2-nitrophenol in the 295-400 nm region and photolysis of 2-nitrophenol at 308 and 351 nm," J. Phys. Chem. A 120, 9958-9967, 2016) are not discussed.

Absorption cross-section data of S. A. Shama ("Vacuum ultraviolet absorption spectra of organic compounds in gaseous and liquid state," PhD Thesis, Faculty of Science (Benha) Zagazig University, Egypt, 1991, http://library.mans.edu.eg/eulc_v5/Libraries/Thesis/BrowseThesisPages.aspx?fn =PublicDrawThesis&BibID=9666196) of mono- and disubstituted aromatics above 170 nm are missing, see  MPIC Spectral Atlas.

Line 8435    correct:    Chen et al. (2011)

Line 8441    delete in title  "for 2-nitrophenol"

Line 8459    correct: e.g.

Line 8461    rewrite:  The quantum yields are based on photolysis rates observed under defined conditions

Not:    the quantum yield based on photolysis rates observed by under defined conditions

Line 8469    correct  …… Peter, P. and Benter

Line 8471    Bardini, P.

Line 8472    correct: Wenger, J. C. and Venables, D. S.,

Line 8655    Bardini, P.

**P30**

Line 8485    add trivial name:  benzaldehyde

Line 8494    align references in table

Line 8500 and further throughout this comment section:
        correct cross-sections (not cross sections)

Line 8509    correct:   ..investigated *at* wavelengths

Line 8520    units are missing:  $cm^2$ $molecule^{-1}$

Line 8525    rewrite: … determined by the "factor analysis method", *where* the
        spectrum obtained is refined and…

Line 8527    correct:  complex

Line 8529    sentence is incomplete:  give wavelength range

Lines 8553 and 8554:    correct units: $cm^2$ $molecule^{-1}$

Line 8557 and 8558:    rewrite: … and those of Zhu and Cronin (2000), which
are significantly lower than other measurements, except at 318 nm where they
are slightly higher.

Line 8561    correct: Chen *et al*….  not …  at al.

Line 8563    rewrite sentence:   …and an offset in their absorption at $\lambda > 380$
        nm, *probably* resulting from a baseline shift *or noise.*

Line 8565    correct: Thiault *et al*. (2004) blue shifted by 4 nm…

Line 8570    remove "at"

Line 8571    correct:  may be unrepresentative…. (delete "in")

Line 8572    insert comma after "Nevertheless"

Lines 8575 to 8589:   enter space between first names of authors

Line  8584   correct: Nozière, B.,  further put first names behind authors

**P31**

Lines 8612 and 8617      correct:  cross-sections (not cross sections)

Lines 8621   correct  3-*methyl*-2-nitrophenol

Line 8628    delete  "for 3-methyl-2-nitrophenol"

Line 8637    correct cross-sections (not cross sections)

Lines 8640 and 8646      correct  3-*methyl*-2-nitrophenol

Line 8641    correct : $cm^2$ $molecule^{-1}$

Line 8644    correct:    Bejan et al. (2006)

Line 8645    correct reference into:    Bejan et al. (2007)

Line 8647    correct       red-shifted

Line 8651    correct       ….Kleffmann, J.,

Lines 8653 and 8656:    remove comma before "and"

Line 8656    enter space between first names of authors

**P32**

Lines 8678 and 8683      correct:  cross-sections (not cross sections)

Line 8693    delete  "for 4-methyl-2-nitrophenol"

Line 8702    correct:  cross-sections (not cross sections)

Line 8707    correct : $cm^2$ $molecule^{-1}$

Line 8708    correct : acetonitrile  (not acetonitryl)

Line 8711    correct:    Bejan *et al.* (2007)

Line 8716    correct ….Kleffmann, J.,

Lines 8718 and 8621      remove comma before "and"

---

## Author Comment (AC1) · 8 Feb 2021

We have added the following in the main text (end of page 3): "The coverage of the evaluation was previously limited to the reactions of $\leq$ C3 organics, selected larger species (n-butane, isoprene, 2-methyl-but-3-en-2-ol and alpha-pinene) and some related oxygenated products (Atkinson et al., 2006). This has been substantially increased to include all C4 alkanes and alkenes, benzene and toluene, monoterpenes and sesquiterpenes, and related oxygenated products. This volume therefore presents evaluations for 97 new reactions, and updates for 25 reactions, of organic species with

four, or more, carbon atoms."

---

## Author Response (AR1)

**Reply to reviewers:**

Thanks to the reviewers for their constructive and valuable comments

**Referee #1:**

This IUPAC evaluation (the eighth in the series) is clearly publishable in ACP. The evaluation panel provides an invaluable service to the atmospheric chemistry community, and the publication of their recommendations in ACP is an entirely appropriate mechanism for broad dissemination of this information. The text is good, and the summary recommendations all seemed correct. I have only one suggestion for improvement. It is obvious that the panel cannot review all C4+ species for which data exist, but can the authors provide a summary in the text of the scope of the species evaluated? Or the rationale for determining which species are included in the review?

- Reply:

We have added the following in the main text (end of page 3):

"The coverage of the evaluation was previously limited to the reactions of $\leq C_3$ organics, selected larger species (*n*-butane, isoprene, 2-methyl-but-3-en-2-ol and $\square$-pinene) and some related oxygenated products (Atkinson et al., 2006). This has been substantially increased to include all $C_4$ alkanes and alkenes, benzene and toluene, monoterpenes and sesquiterpenes, and related oxygenated products. This volume therefore presents evaluations for 97 new reactions, and updates for 25 reactions, of organic species with four, or more, carbon atoms."

**Referee #2:**

As stated in the abstract, this is the eight overview article published in ACP on the detailed evaluation work carried out by the IUPAC Task Group on Atmospheric Chemical Kinetic Data Evaluation, focusing on the gas phase reactions of organic species with four, or more, carbon atoms. The work carried out by this group is of critical importance to the atmospheric chemistry community, where these evaluations drive our current understanding of tropospheric and stratospheric chemistry represented in the chemical mechanisms that underpin all air quality and climate science and policy models. This overview article summarises the datasheets and recommendations on the reactions of selected tropospherically important ≥ C4 closed-shell organic species with HO and NO3 radicals, and their photolysis. The datasheets and recommendations discussed are available, along with recommendations of other atmospherically important gas, heterogeneous and aqueous phase reactions on a dedicated webpage, which is updated periodically. The article itself is a well written and clear summary which certainly should be published in ACP. One suggestion is that it would be useful to the reader if an additional table is added which summarises the range of VOCs covered, e.g. how many alkanes, alkenes, aromatics, oxygenated species, organic nitrates and nitro compounds, etc. . ., with the supplementary datasheets logically ordered as appropriate.

- Reply:

In addition to the clarification of scope outlined in the response to Referee #1, we have added the following in the abstract of the main text:

"The present work is a continuation of Volume II (Atkinson et al., 2006) with new reactions and updated datasheets for reactions of HO (77 reactions) and $NO_3$ (36 reactions) with $\geq$ C4 organics including alkanes, alkenes, dienes, aromatics, oxygenated, organic nitrates and nitro compounds in addition to photochemical processes for 9 species."

**Referee #3:**

This paper presents the eighth evaluation of kinetic and photochemical data by the IUPAC Task Group on Atmospheric Chemical Kinetic Data Evaluation. The work performed by this evaluation panel is of prime importance for the scientific community of atmospheric sciences, in particular for the development of chemical schemes in atmospheric models but also for lab and field experimental studies. Recommendations are well justified in the supplement document and appear reasonable. This paper, which complements well the online database, is very useful for users as it includes a theoretical section on the kinetic treatment of the various types of atmospheric reactions in the gas phase. It also includes a summary of recommended kinetic parameters as well as detailed information upon which these recommendations are made. This paper is clear and well written. For rate constants that have been subject to re-evaluation, I would suggest to provide in the paper, or in the interactive database, the former value (for example in brackets). This could be very useful for the users as it would improve the traceability of the changes made. As a recommendation, this is not a critical point for the publication of the paper which can be published in its current state. However, I think that it could be useful for the users. In conclusion, I fully support the publication of this paper.

- Reply:

We thank the reviewer for the suggestion of providing former values in brackets. We are working on updating the website to enable automatic generation of summary tables and plan to highlight changes to previous recommendations as the reviewer suggests. Most of the presented data sheets were not previously in Vol. II. This is therefore the first time the preferred values have been published in the peer-reviewed literature.

**Referees #4 (Cornelius Zetzsch and Geert Moortgat):**

This current article is a supplement to the "Evaluated kinetic and photochemical data for atmospheric chemistry, Volume II; gas-phase reactions of organic species" (Atkinson et al., 2006). It updates existing data sheets, presents several new ones and continues to be a valuable tool for the scientific community. The kinetic and photochemical datasheets of the present work are (or are about to be) accessible in the internet (www.iupac.pole-ether.fr). The manuscript is clearly publishable in ACP, where the readers may take the opportunity to add hints at errors and other work for consideration (further to corresponding with the lead author or any other member of the task group). The article starts with an introduction into the series with volumes on   Ox, HOx, NOx, and SOx species   organic species (the topic of the present work) inorganic halogens organic halogen species   heterogeneous reactions on solid substrates   heterogeneous reactions with liquid substrates  and, in preparation, reactions of organic species with ozone and chemistry of Criegee intermediates It continues to present a guide to the datasheets with precise information on the equations of the functions employed and to the assignment of uncertainties in the recommendations of rate coefficients and Arrhenius parameters for room temperature and below. Moreover, the supplement shows Arrhenius diagrams for more than 40 molecules (how many alkanes, olefins and terpenoids, alcohols, aldehydes, acids, nitro compounds, aromatics?) and diagrams of UV absorption cross- sections for 8 molecules. You may wish to list the

Arrhenius diagrams with trivial names of the molecules on page 2, line 69 and might point out features of positive and negative temperature dependence (and convex and concave curvature), to be observed in the supplement.

**General and specific comments and technical corrections:**

P. 1, L. 25: The abstract might mention that the present work is a continuation of Volume II (Atkinson et al., 2006) with new (which molecules?) and updated data sheets for Appendix A2 (HO reactions), A3 (NO3 reactions) and A8 (photochemical processes).

Done

P.1, l. 26: Have all of the data sheets of the present supplement been evaluated in 2019 or should the date be given individually for each data sheet or should the IUPAC website be consulted?

The latest evaluation dates are now given in the datasheets

P. 3, line 73: gas‑phase might be cancelled (see title)

Done

P. 4: There are further abbreviations in the supplement, which could be listed, such as API, FEP, CEAS, IBBCEAS, ToF, EUPHORE

Abbreviations: API, FEP, CEAS, IBBCEAS, ToF added

P. 9, l. 285: Are there appropriate, obvious examples for combination reactions without a barrier?

In this general section we would like to maintain consistency with previous IUPAC publications and prefer not to give examples.

P. 12, l. 386: In which cases is H2O a particularly efficient third body?

In this general section we would like to maintain consistency with previous IUPAC publications and prefer not to give examples.

P.12, l. 398: Photochemical transitions or processes? The data sheets comprise cases, where available UV spectra have not been discussed down to 170 nm, where UV spectra for, e.g., 2‑nitrophenol are available (http://satellite.mpic.de/spectral_atlas).

Taken into account:

- Processes is used and 170 nm replaced by 185 nm

P. 14, l. 460: "Unfortunately" or better: "On the other hand" or "Moreover" (?).

"However" is used

This situation stimulates further investigation with improved techniques, wider ranges of concentration, total pressure and temperature, or detailed consideration of the tail of non‑exponential decays as biexponential, triexponential, multiexponential or mixed with second order components

P. 14, l. 463‑466: Some more reasons for such differences of more than a factor of two could be listed here, such as insufficient purity of compounds, unknown/unexpected impurities, wall adsorption and decomposition, insufficient time resolution for the initial elementary step, opposing and parallel reactions or unexpected decomposition products from the primary product.

The following has been added:

"The differences between various measurements could be due to multiple reasons, such as insufficient purity of compounds, unknown/unexpected impurities, wall adsorption and decomposition, insufficient time resolution for the initial elementary step, opposing and parallel reactions or unexpected decomposition products from the primary product."

P. 15, line 473: "scarcity of reliable data" => could you delete "reliable" (?)

Done

P. 15, line 483: IUPAC. For the => IUPAC for the

Done

P. 16. line 524: "J. Phys. Chem. Ref. Data" occurs twice

The second has been deleted

P. 17, line 550: One might expect the series of evaluations by Atkinson (monographs) and by Calvert et al., most of which are mentioned in the supplement in several datasheets, to be cited here as well.

Our introductory text gives an overview of the work of the IUPAC panel but does attempt to provide a comprehensive listing of previous evaluations / databases of kinetic data.

P. 19 – 22: Overall or overall?

Done

P. 22: Table 1 should end with the NO3 reactions and the footnote, and the photochemical reactions should be a separate Table 2 (with different entries, such as absorption cross‐sections and quantum yields?).

The photochemical data is not parameterised in a systematic manner that can be easily tabulated in a short form for all of the datasheets. Therefore, we have chosen to simply list the processes covered and name the datasheet in which the details about the absorption spectrum and the quantum yields can be found.

**Supplement**

**General remarks on the text**

It occurs in several places of the supplement (unlike the main manuscript) that the list of authors in the references does not end with a colon before the journal follows (in the photolysis section mainly).

Corrected

Exponents and their numerals should not appear in separate lines.

Corrected

The dates of final evaluation (and recommendation) should be mentioned in the heading of every data sheet, like those available on the website.

Done

Unlisted authors (et al.), isomers cis, trans, H‐, n‐ and i‐ should be marked by italic fonts in the whole text, in the tables and in the comments.

isomers cis, trans, H-, n- and i-are in italic now (al. not modified)

The date of the IUPAC recommendations (IUPAC and year instead of Arrhenius fit (or biexponential fit on page 212) should appear in the figures

For anyone interested, information concerning the date of the IUPAC evaluation and the last change in preferred values can be readily accessed via the website for any particular datasheet.

The use of upper and lower case letters in headings, footnotes and figure captions is inconsistent.

Corrected

Pages 1 – 3 Some simple structures of aliphatics are missing, and trivial names of several molecules could be added as explanation in brackets, even for catechol.

Chemical names are given in the table. The names and structures are given in datasheets

Tables on Pages 5 (n‐butane), 141 (benzene), 147 (toluene): Recent work exists on relative rate constants for 17 hydrocarbons, including n‐butane, benzene and toluene. These have been investigated at 248 and 288 K in a smog chamber by Han et al. (L. Han, F. Siekmann, C. Zetzsch: Atmosphere 9, 320, 2018, doi:10.3390/atmos9080320) in reasonable agreement with the IUPAC recommendations.

Thanks to the reviewer, the above data have been added.

Page 141, Line 4110:    => pressure. The non‐exponential decays were shown to be biexponential by Wahner and Zetzsch (1983), Knispel et al. (1990), and Bohn and Zetzsch (1999), according to the analytical solution of the differential equation system, where the back‐decomposition of the HO‐benzene adduct, leads to an equilibration of OH with the adduct, considering the abstraction channel as irreversible loss process of OH and the adduct.

The suggestion was added to the end of comment (f):

(f)    Rate coefficients were measured over the pressure ranges 67-173 mbar (50-130 Torr) of Ar diluent (Wahner and Zetzsch, 1983) and 33-666 mbar (25-500 Torr) of He diluent (Rinke and Zetzsch, 1984), with a slight decrease in rate coefficient being observed below 133 mbar (100 Torr) pressure in both cases. The cited rate coefficients are at 133 mbar pressure. *"The non-exponential decays were shown to be biexponential by Wahner and Zetzsch (1983), Knispel et al. (1990), and Bohn and Zetzsch (1999), according to the analytical solution of the differential equation system, where the back-decomposition of the HO-benzene adduct, leads to an equilibration of OH with the adduct, considering the abstraction channel as irreversible loss process of OH and the adduct."*

**Figures**

Unlisted authors (et al., sometimes those who spent their time performing the experiments or improving the equipment) are missing in several figures (especially the NO3 section), in some instances the second author as well.

The present labelling system enables the responsible authors (listed in full in the references) to be identified unambiguously and avoids cluttering the Figures with too much information. All authors are listed correctly in the reference list, that is fine.

Not in all cases are the rate coefficients and Arrhenius expressions derived by other authors listed in the tables and shown in the graphs (this would help the reader even if no IUPAC recommendation is made for a single study of the temperature dependence). This holds even more so if an expression

derived by other authors is adopted by IUPAC (e.g. n‑butane, adopted from Donahius and Clarke, 2004).

Adding all reported Arrhenius expressions to the Figures would result (in many cases) in an unreadable graph and we prefer not to do this. Where available, we have listed the reported Arrhenius expressions to the tables in each data-sheet.

P. 8 (n‑Butane): correct Clark into Clarke, add tick marks or gridlines of the logarithmic scale. Gridlines might be added to all figures

Clark corrected into Clarke. No change in the Figures made

P. 17 (Isoprene) The reaction HO + Isoprene does not appear in the figure as a label. In the same fashion, labels with the reactions are missing on pages 22, 28, 79, 118, 122, 125, 128, 130, 132, 215, 217, 224, 226, 231, 233, 235, 237.

Done for all

Delete black boxes around the reaction labels on pages 83, 86 and 197

Done for all

Except for a few aromatics, the abstraction channel has not been shown in the Arrhenius diagrams. Equilibration of OH and NO3 with the adduct may cause the negative activation energy for olefinic compounds and a concave curvature at high temperatures. Have available high‑temperature data or non‑exponential decays at intermediate temperatures been considered in all cases?

In the Figures, we plot the overall rate constant. The (temperature dependent) fractional contribution of the abstraction and addition channels is parameterised in the table of preferred valued.

**Technical corrections**

Line 2853: at al.  => *et al.*

Not changed

Line 2969: at al.  => *et al.*

Not changed

Line 3235: at al.  => *et al.*

Not changed

Lines 3450 and 6418: The asterisk might be explained by a footnote, or one might ask the authors for the correct temperature

A footnote was added under the tables

   *) the experimental temperature was "room temperature" which we list as 298K.

Line 4095: non‑exponential  => biexponential: Done

Line 4117: non‑exponential  => biexponential: Done

Line 4194: non‑exponential  => biexponential: Done

Line 4290: non‑exponential  => biexponential: Done

Line 4405: non‑exponential => biexponential: Done

Line 4211: Somerlade => Sommerlade: Done

Line 4789: non‑exponential => biexponential: Done

Line 5656: to appear => add year of appearance (2011): Done

The following suggestions have been taken care of:

Line 5834: => over 2‑methylpropene

Line 5838: => over 2‑methylpropene

Line 5840: => of 2‑methylpropene

Line 6153: => Bunsenges.

Lines 6186, 6286, 6525, 6600, 6763, 6836, 6968: NO3. So that…. => NO3, so that

Line 6436: 1 bar

Lines 6542, 6781, 6861, 6922: accurate => absolute

Line 6746: Bunsen‑Ges. => Bunsenges.

Line 6876: Bunsen‑Ges. => Bunsenges.

Line 6879: cancel VOC55

Line 7030: 8‑methylene..

Line 7254: 7‑tetra

Line 7290: a‑dimethyl

Line 7329: a‑hexa..

Line 7463: was obtained.

Line 7533: agreement, and => agreement and (?)

Line 7566: RRGC => RR‑GC

Line 7617: butene.

Line 7622: 5‑methyl⋯

Line 7679: butene.

**Photochemistry section**

**General comments**

**1  Selection criteria**

In this supplement of Volume II (Atkinson et al., 2006), 8 organic species with four, or more, carbon atoms ($\geq$ C4) were evaluated.  What are the criteria of the selection of those eight organic species? More photolysis studies of organic species ($\geq$ C4) have been published in the literature, which could have been evaluated in this supplement.

Examples are: n‑butanal and n‑pentanal (Tadic et al., 2001a), n‑hexanal (Tadic et al., 2001b), n‑heptanal (Tadic et al., 2002), n‑octanal (Tadic et al., 2011), trans‑crotonaldehyde (Magneron et al., 2002) and methyl ethyl ketone (Nádasdi et al., 2010)

**References**

Magneron, I., Thévenet, R., Mellouki, A., Le Bras, G., Moortgat, G. K. and Wirtz, K.: J. Phys., Chem., A, 106, 2526 (2002).

Nádasdi, R., Zügner, G. L., Farkas, M., Dóbé, S., Maeda, S. and Morokuma, K.: Chem. Phys. Chem., 11, 3883 (2010).

Tadic, J., Juranic, I. and Moortgat, G. K.: J. Photochem. Photobiol. A: Chem., 143, 169 (2001a).

Tadic, J. M., Juranic, I. and Moortgat, G. K.: Molecules, 6, 287 (2001b).

Tadic, J. M., Juranic, I. O. and Moortgat, G. K.: J. Chem. Soc., Perkin Trans., 2, 135 (2002).

Reply:
A criterion that we generally use is that the oxygenate is a product of a hydrocarbon or other species in the evaluation. A large number of species has not been evaluated so far. This will be done in the future. However, we have updated the datasheet for butanone (P8) and added it to the present volume.

**2  Presentation**

a) In the Summary page (p 271, line 8010) one product channel is given, which is not correct. One ought to replace this by "products", as is shown in the datasheets, or add the other product channels

Corrected

b) The head texts of the current datasheets are different from those presented in Volume II (Atkinson et al., 2006). The text that appeared on the website   (iupac.pole‑ether.fr) contains additional information, such as the update date

Head texts of the datasheets completed with the dates

c) In the title molecule, it would be advisable to add the trivial name d) The presentation section starts with "Primary photochemical transitions" However, in older datasheets of Volume IV (Atkinson et al., 2008), this title was named "Primary photochemical processes". Would this title be more appropriate?

e) In all photochemical datasheets, the substances appear above the figure

The substances are now in the Figures boxes

**3  Technical corrections**

A) Throughout the text in Volumes II and VIII, the references are not presented uniformly. It is advisable to use the same citation style throughout the manuscript, including the supplement

The references are presented uniformly

**B) Units.**

The text should use for the cross-section units $cm^2$ $molecule^{-1}$ and not $cm^2 molecule^{-1}$ nor $cm^2/molecule$

$cm^2$ $molecule^{-1}$ is used

Additional figures of the spectra on a logarithmic scale (or a reference to the Spectral Atlas, where these are an option) might be useful.

Term symbols of the transitions (absorption bands displayed in the figures) and rough estimates of the oscillator strengths would be useful and a key to the photolytic processes.

Our emphasis is on cross-sections, and quantum yields (which are needed to evaluate the lifetime and role of any trace gas in the atmosphere) and not spectroscopic details of the excited states involved.

C) Individual remarks

**P23 :** all suggestions taken into account

Line 8022 add trivial name:  biacetyl

Line 8027 align reaction products

Line 8063 correct p = ∞      into   p → ∞

Line 8071     remove one comma: Horowitz et al. (2001), which are….

Line 8092 correct:   Barnes, I. and Becker K. H.,

Line 8094 correct:  Calvert, J. G. and Pitts Jr., J. N.,

Line 8098 correct: Ravishankara, A. R. and Burkholder, J. B.,

Line 8104 correct: Absorption spectrum of biacetyl

**P24** all suggestions taken into account

Line 8111   add trivial name:  i‑butyraldehyde

Line 8114 align reaction products

Line 8130   correct: with a resolution

Line 8135 remove comma:   … determined from measurements…

Line 8153/54 rewrite:  …except for very slightly at 330.5 nm . >>>> ….except at 330.5 nm, where a minimal pressure dependence was observed.

Line 8174 replace:… better than 4 %... by … smaller than 4%

Line 8197 correct: …Francisco, J. S.: J. Phys. Chem. A, …. 2002.

Line 8200 correct:  Calvert, J. G. and Pitts Jr., J. N.,

Line 8206   correct:   Absorption spectrum of i‑butyraldehyde

**P26** all suggestions taken into account

Line 8213 add trivial name.

   Note: two different names appear in the text:     butenendial and butene‑2‑dial

It is assumed that butane-2-dial is correct, and should be corrected throughout the text.

Line 8213 move arrow, and align product channel numbers

Line 8219 correct reference:  Hufford et al., 1952.

Line 8222 the transitions "cis‑/trans & trans/‑cis" should be in italics font

Line 8226 the Comments a, b, c, d, and e, are erroneously labeled a, b, c, c,and d

Lines 8230, 8232, 8240   correct units to:  $cm^2$ molecule$^{-1}$

Line 8232 move right bracket: purified (crystalline) fumaric dialdehyde

Lines 8232 and 8239   correct: cross‑sections (not cross sections)

Line 8242 insert commas: at 193 nm, HCO produced by the Cl + HCHO reaction, was…

Line 8243 delete " in"

Line 8247 correct: "was" into "were"

Line 8253 correct:  assigned

Lines 8255 and 8282: add year (1994) of reference

Line 8257 change temperature to 298 K

Line 8271 correct  Fig 1    to    Fig. 1

Line 8278 insert comma after …limits >>>… limits,

Line 8287   correct:  ….  Barnes, I., Becker, K. H. and Wiesen, P., Environ. Sci. Technol.

Line 8289 correct:  …Chem. Phys. Lett.

Line 8242 enter space between first names of authors

Line 8305 correct:  Absorption spectrum…

**P27** all suggestions taken into account

Line 8311   Note: three different names appear in the text:

   4-oxo-pent-2-enal , 4‑oxo‑2‑pentenal and 4‑oxo‑penten‑2‑dial (see figure caption)

Which is correct?, This should be corrected throughout the text.

Line 8313 align product channels

Line 8322 the transitions "*cis‑trans & trans‑cis*" should be in italics

Line 8328 and further throughout this comment section:

         correct cross‑section (not cross section)

Lines 8352 and 8354 correct:  5-methyl-3H-furan-2-one,    not     5-Methyl-3H-furan-2-one

Line 8358 add year (1994) of reference

Line 8370 change temperature to 298 K

Line 8378 correct:   ⋯ study of the gas‑phase⋯

Line 8383   enter space:  193 nm

Line 8395 insert comma after:  However,…

Line 8397 correct Bierbach et al. (1994)

Line 8401 add year (1994) of reference

Line 8411 correct figure caption: .. spectrum of  [enter correct name]

**P28**

Line 8421:

Absorption cross‑section data and quantum yields of M. Sangwan and L. Zhu ("Absorption cross sections of 2‑nitrophenol in the 295‑400 nm region and photolysis of 2‑nitrophenol at 308 and 351 nm," J. Phys. Chem. A 120, 9958‑9967, 2016) are not discussed.

Absorption cross‑section data of S. A. Shama ("Vacuum ultraviolet absorption spectra of organic compounds in gaseous and liquid state," PhD Thesis, Faculty of Science (Benha) Zagazig University, Egypt, 1991, http://library.mans.edu.eg/eulc_v5/Libraries/Thesis/BrowseThesisPages.aspx?fn=PublicDrawThesis&BibID=9666196) of mono‑  and disubstituted aromatics above 170 nm are missing, see  MPIC Spectral Atlas.

Reply:

The data from M. Sangwan and L. Zhu have been added to the datasheet P28. In addition P31 and P32 have been updated. *However, we do not use non-peer reviewed sources*

Line 8435 correct:    Chen et al. (2011)

Line 8441 delete in title   "for 2‑nitrophenol"

Line 8459 correct: e.g.

Line 8461 rewrite:  The quantum yields are based on photolysis rates observed under defined conditions Not:  the quantum yield based on photolysis rates observed by under defined conditions

Line 8469 correct  …… Peter, P. and Benter

Line 8471    Bardini, P.

Line 8472   correct: Wenger, J. C. and Venables, D. S.,

Line 8655     Bardini, P.

**P30** all suggestions taken into account

Line 8485 add trivial name:  benzaldehyde

Line 8494 align references in table

Line 8500 and further throughout this comment section:  correct cross‑sections (not cross sections)

Line 8509 correct:   ..investigated at wavelengths

Line 8520    units are missing: cm2 molecule‑1

Line 8525 rewrite: … determined by the "factor analysis method", where the spectrum obtained is refined and…

Line 8527   correct:  complex

Line 8529 sentence is incomplete:  give wavelength range

Lines 8553 and 8554:   correct units: cm2 molecule ‑ 1

Line 8557 and 8558:     rewrite: … and those of Zhu and Cronin (2000), which are significantly lower than other measurements, except at 318 nm where they are slightly higher.

Line 8561 correct: Chen et al….  not …  at al.

Line 8563 rewrite sentence:    …and an offset in their absorption at $\lambda > 380$ nm, probably resulting from a baseline shift or noise.

Line 8565 correct: Thiault et al. (2004) blue shifted by 4 nm…

Line 8570 remove "at"

Line 8571 correct:  may be unrepresentative…. (delete "in")

Line 8572 insert comma after "Nevertheless"

Lines 8575 to 8589:   enter space between first names of authors

Line 8584   correct: Nozière, B.,  further put first names behind authors

**P31** all suggestions taken into account

Lines 8612 and 8617   correct:  cross ‑ sections (not cross sections)

Lines 8621   correct  3 ‑ methyl ‑ 2 ‑ nitrophenol

Line 8628 delete   "for 3 ‑ methyl ‑ 2 ‑ nitrophenol"

Line 8637 correct cross ‑ sections (not cross sections)

Lines 8640 and 8646 correct  3 ‑ methyl ‑ 2 ‑ nitrophenol

Line 8641 correct : cm2 molecule-1

Line 8644   correct:    Bejan et al. (2006)

Line 8645 correct reference into:   Bejan et al. (2007)

Line 8647 correct red ‑ shifted

Line 8651   correct  ….Kleffmann, J.,

Lines 8653 and 8656:   remove comma before "and"

Line 8656 enter space between first names of authors

**P32** all suggestions taken into account

Lines 8678 and 8683   correct:  cross ‑ sections (not cross sections)

Line 8693 delete  "for 4-methyl-2-nitrophenol"

Line 8702 correct:  cross‑sections (not cross sections)

Line 8707 correct : $cm^2$ $molecule^{-1}$

Line 8708 correct : acetonitrile  (not acetonitryl)

Line 8711 correct:    Bejan et al. (2007)

Line 8716 correct ….Kleffmann, J.,

Lines 8718 and 8621 remove comma before "and"